# Exosomes harbor B cell targets in pancreatic adenocarcinoma and exert decoy function against complement-mediated cytotoxicity

Michela Capello[1], Jody V. Vykoukal[1,2], Hiroyuki Katayama [1], Leonidas E. Bantis[3,4], Hong Wang[1], Deepali L. Kundnani [1], Clemente Aguilar-Bonavides[3], Mitzi Aguilar[1], Satyendra C. Tripathi [1], Dilsher S. Dhillon[1], Amin A. Momin[1], Haley Peters[5], Matthew H. Katz[6], Hector Alvarez[7], Vincent Bernard[7], Sammy Ferri-Borgogno[7], Randall Brand[8], Douglas G. Adler[9], Matthew A. Firpo [10], Sean J. Mulvihill[10], Jeffrey J. Molldrem[5], Ziding Feng[3], Ayumu Taguchi[11], Anirban Maitra[7,11] & Samir M. Hanash [1,2,11]

Although B cell response is frequently found in cancer, there is little evidence that it alters tumor development or progression. The process through which tumor-associated antigens trigger humoral response is not well delineated. We investigate the repertoire of antigens associated with humoral immune response in pancreatic ductal adenocarcinoma (PDAC) using in-depth proteomic profiling of immunoglobulin-bound proteins from PDAC patient plasmas and identify tumor antigens that induce antibody response together with exosome hallmark proteins. Additional profiling of PDAC cell-derived exosomes reveals significant overlap in their protein content with immunoglobulin-bound proteins in PDAC plasmas, and significant autoantibody reactivity is observed between PDAC cell-derived exosomes and patient plasmas compared to healthy controls. Importantly, PDAC-derived exosomes induce a dose-dependent inhibition of PDAC serum-mediated complement-dependent cytotoxicity towards cancer cells. In summary, we provide evidence that exosomes display a large repertoire of tumor antigens that induce autoantibodies and exert a decoy function against complement-mediated cytotoxicity.

---

[1] Department of Clinical Cancer Prevention, The University of Texas MD Anderson Cancer Center, Houston, TX 77030, USA. [2] The McCombs Institute for the Early Detection and Treatment of Cancer, The University of Texas MD Anderson Cancer Center, Houston, TX 77030, USA. [3] Department of Biostatistics, The University of Texas MD Anderson Cancer Center, Houston, TX 77030, USA. [4] Department of Biostatistics, University of Kansas Medical Center, Kansas City, KS 66160, USA. [5] Department of Stem Cell Transplantation and Cellular Therapy, The University of Texas MD Anderson Cancer Center, Houston, TX 77030, USA. [6] Department of Surgical Oncology, The University of Texas MD Anderson Cancer Center, Houston, TX 77030, USA. [7] Department of Pathology, The University of Texas MD Anderson Cancer Center, Houston, TX 77030, USA. [8] Division of Gastroenterology, Hepatology and Nutrition, University of Pittsburgh, Pittsburgh, PA 15232, USA. [9] Department of Internal Medicine, University of Utah School of Medicine, Salt Lake City, UT 84132, USA. [10] Department of Surgery, University of Utah School of Medicine, Salt Lake City, UT 84132, USA. [11] Department of Translational Molecular Pathology, The University of Texas MD Anderson Cancer Center, Houston, TX 77030, USA. Correspondence and requests for materials should be addressed to S.M.H. (email: shanash@mdanderson.org)

B-cell-associated autoimmune response is found in most tumor types and is evidenced by the production of auto-antibodies against tumor-associated antigens (TAAs)[1]. The production of autoantibodies may precede disease symptoms by months or years[2]. As a result, detection of tumor-associated autoantibodies in the circulation represents a feasible approach for cancer-early detection[3,4]. The process through which TAAs are recognized by the immune system and thereby trigger a humoral response is not well delineated. TAAs are not restricted to proteins carrying mutations and are often represented by proteins with no discernable alterations in their structure. Rather, altered localization or post-translational modifications are found to elicit production of autoantibodies[5]. The functional significance of a humoral immune response in cancer is not clear as there is inconsistent evidence that it alters tumor development or progression.

Exosomes are 30–150 nm diameter extracellular vesicles (EVs) that arise by specific endosomal biogenesis pathways[6]. Exosomes harbor a diverse repertoire of molecular cargo that includes proteins, RNA, and DNA derived from their originating cells and that are shielded from degradation in the circulation[7–9]. EVs have emerged as mediators of intercellular communication and potential reservoirs of biomarkers[10–12]. Exosomes also have important roles in immune response. Tumor-derived exosomes containing TAAs can transfer MHC-peptide complexes as well as whole antigens to dendritic cells (DCs) for processing and cross-presentation to tumor-specific T lymphocytes[13]. There is also evidence that tumor-derived exosomes may exert a suppressive effect on both adaptive and innate antitumor responses[14].

Through comprehensive proteomic analyses of plasma-derived circulating antigen-antibody complexes and of cancer cell line- and plasma-derived exosomes, we have investigated the contribution of tumor-associated exosomes to the repertoire of autoantibodies in pancreatic adenocarcinoma. Here, we demonstrate that tumor-derived exosomes are bound to circulating immunoglobulins in the plasma and that in particular the surface membrane of tumor exosomes displays a large repertoire of TAAs that are targets of autoantibodies. We provide evidence of a decoy function of exosomes that attenuates complement-mediated cytotoxicity directed at tumor cells.

## Results

### Exosomes are bound to immunoglobulins in PDAC plasmas.

We performed in-depth proteomic profiling of immune complexes derived from plasma samples of patients with pancreatic ductal adenocarcinoma (PDAC). Circulating immunoglobulins (Igs) were isolated from the plasma by affinity-capture and Ig-bound proteins were identified by liquid chromatography-tandem mass spectrometry (LC-MS/MS) (Fig. 1a). Analyses were performed using plasma sample pools from PDAC patients, which were compared to pools of matched healthy subjects, benign pancreatic cyst patients, and patients with chronic pancreatitis (cohort #1 and #2; Fig. 1b and Supplementary Table 1). In total, 308 proteins were identified in the Ig-bound fractions with at least five normalized MS2 spectral counts (Supplementary Data 1). Ninety-two proteins were selected from this list based on the following criteria: (i) a case-to-matched control average MS2 count ratio of 1.5 or greater; and (ii) confirmed expression of the corresponding genes in a panel of 11 PDAC cell lines, as well as in The Cancer Genome Atlas (TCGA) PDAC dataset ($n = 112$ patients). Fifty-nine of the selected Ig-bound proteins (64%) were reported as significantly overexpressed in PDAC compared to normal adjacent tissue in two or more of the eight PDAC gene expression datasets available in the Oncomine database (www.oncomine.org), (Supplementary Tables 2 and 3). Among the 92

proteins, molecules annotated to be involved in vesicular trafficking and biogenesis such as RAN, ARF6, Endofin (ZFYVE16), TMEM175, ATP9B, and ITPR2, were identified (Supplementary Tables 2 and 3). MetaCore Gene ontology localization analysis indicated a statistically significant enrichment of EV/exosome- and endosome-associated proteins within the selected Ig-bound proteins. The top four predicted localizations were extracellular exosome, EV, extracellular organelle, and vesicle exhibiting a $p < 8.44 \times 10^{-11}$ and an false discovery rate (FDR) $< 8.29 \times 10^{-9}$ (Fig. 1c). High-stringency low pH washing conditions indicated negligible non-specific binding of exosomes to the protein A/G columns that were applied for the affinity capture of circulating Ig complexes (Supplementary Fig. 1).

The finding that Ig-bound proteins were enriched in exosome-associated proteins suggested the possibility that intact exosomes from PDAC patients may be bound to Ig in circulation. Affinity-purified plasma Ig fractions from PDAC patients and matched healthy controls ($n = 7$ per group, cohort #3; Supplementary Table 1) were subjected to exosome isolation using ultracentrifugation-based flotation through a single-step density-overlay ($\rho = 1.14 \text{ g mL}^{-1}$) to separate vesicle-associated from free proteins (Fig. 2a). Nanoparticle-tracking analysis revealed the presence of vesicles in the harvest fraction with an average diameter of $110 \pm 16$ nm, consistent with exosomes (Fig. 2b). No significant difference in particle size between PDAC patients and matched controls was observed (Fig. 2b and Supplementary Fig. 2). In agreement with the nanoparticle-tracking data, transmission electron microscopy (TEM) analyses confirmed enrichment of typical exosome-sized (30–200 nm diameter) vesicles in the low-density harvest fraction (Fig. 2c). Both analyses demonstrated statistically significant enrichment of exosomes in the Ig-bound fractions of PDAC patients compared to the matched controls ($p < 0.05$, one-sided unpaired $t$-test; Fig. 2d). The identity of the recovered vesicles as exosomes was further confirmed through positive detection by flow-cytometry analysis of the isolated vesicles coupled to latex beads of the exosome markers CD63 and TSG101[15]. Additionally, a high level of human IgG was also evident in the recovered exosomes, as revealed by flow-cytometry analysis (Fig. 2e).

To further confirm binding of Igs to circulating exosomes from PDAC patient plasmas, exosomes were isolated from the plasma of six PDAC patients and six matched healthy subjects using the ultracentrifugation-based flotation approach (cohort #4; Supplementary Table 1, Fig. 3a, and Supplementary Fig. 3a). TEM and nanoparticle-tracking analyses confirmed enrichment of typical exosome-sized vesicles in the low-density harvest fraction (Supplementary Fig. 3b and 3c). These plasma-derived exosomes were confirmed by immunoblot to be positive for the markers CD9, CD63, and TSG101 (Supplementary Fig. 3d). Notably, flow-cytometry analysis revealed high levels of human IgG in the isolated exosome-enriched low-density fraction (Fig. 3b). LC-MS/MS was applied to PDAC patient and control plasma-derived exosomes. Proteomic profiling quantified proteins involved in exosome biogenesis, vesicular trafficking, and cytoskeletal regulation, as well as exosome cargo enriched proteins annotated in the Exocarta database (www.exocarta.org) (Supplementary Table 4). Six Ig chains from different subclasses, were identified in the exosome fraction that exhibited >3.5-fold increase in the relative spectral abundancy (MS1 peptide ion intensity) in cases versus matched controls after normalization on the total number of exosomes isolated in each sample (Fig. 3c).

### PDAC cell line exosome surface is enriched in tumor antigens.

Next, we examined whether antigenic proteins that induce autoantibodies in PDAC patients were enriched on the exosome

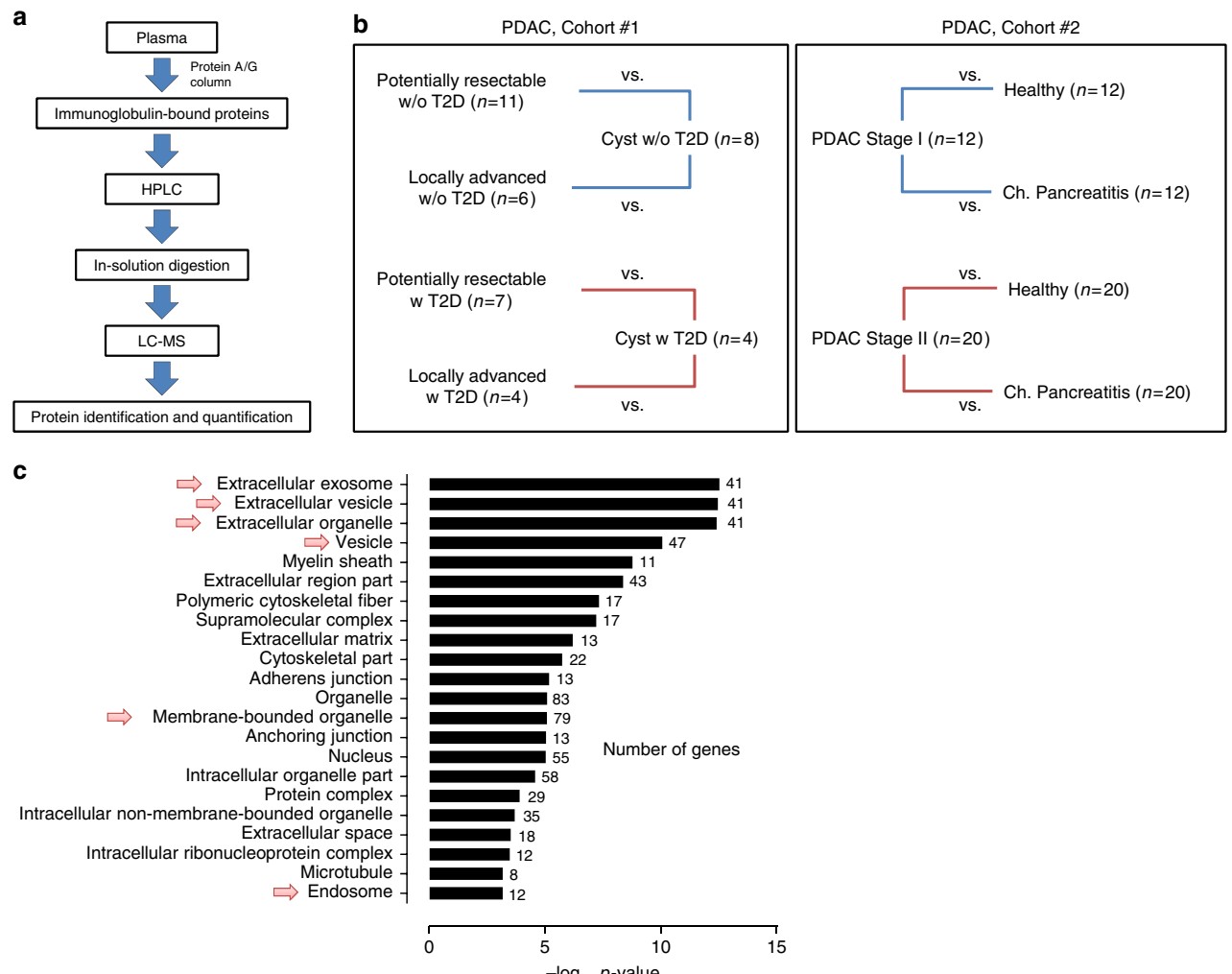

**Fig. 1** Proteomic profiling of proteins bound to circulating immunoglobulins. **a** Schematic of the work-flow used for the proteomic analysis of Ig-bound proteins in the plasma of PDAC cancer patients. **b** Schematic of PDAC cohorts, which underwent MS analysis of Ig-bound proteins. In cohort #1, pools of potentially resectable or locally advanced PDAC patient plasma were compared to pools of benign pancreatic cyst patient samples, matched based on type 2 diabetes (T2D) status. In cohort #2, pools of stage I or stage II PDAC patient plasma were compared to pools of matched healthy subject or chronic pancreatitis patient samples. (**c**) GO MetaCore localization analysis of the proteins identified at higher levels in the Ig-bound fraction of cancer cases compared to matched controls in the PDAC cohorts. The number of genes in the input list belonging to each specific localization is listed. Arrows indicate exosome, vesicle, and endosome localizations

surfaceome of PDAC cell lines. Comprehensive proteomic profiling of individual cell compartments by LC-MS/MS was applied to six human PDAC cell lines (CFPAC-1, HPAF-II, SU.86.86, Panc 03.27, MIA PaCa-2, and PANC-1) and to exosomes isolated from the conditioned media of these lines. Exosomes were isolated using a standard ultracentrifugation method[15]. Nanoparticle-tracking analyses revealed exosomes with an average size of $82 \pm 3$ nm and TEM revealed cup-shaped vesicles in the range of 30–200 nm in diameter (Supplementary Fig. 4a and 4b). The isolated exosomes were positive for the exosomal markers CD63, TSG101, CD9, and Alix as evaluated by immunogold TEM, western-blot, and flow-cytometry analyses (Supplementary Fig. 4a, 4c, and 4d). Immunoblot analysis also revealed specific enrichment of exosomal markers in the exosome isolates compared to the total protein extract of the cell lines from which the vesicles were derived (Supplementary Fig. 4c).

Cell and exosome surface proteins were tagged with biotin and separated from their respective cytoplasmic and cargo compartments by affinity capture to monomeric avidin. Proteomes from total cell extract (TCE), cell surface, total exosome extract (TEE),

exosome surfaceome, and exosome cargo were fractionated at an intact protein level and subjected to trypsin digestion and MS-based analysis (Fig. 4a). In total, we identified 4125 proteins that were detected with ≥5 normalized MS2 counts. Relative to their respective cell line proteomes, the exosome isolates exhibited enrichment in previously described exosome-associated proteins (CD81, CD9, FLOT1, FLOT2, PDCD6IP, SDCBP, and TSG101) and underrepresentation of endosomal proteins not expected to be highly expressed in exosomes (CANX, CYC1, GOLGA2, and HSP90B1)[16] (Fig. 4b).

Proteomic analysis yielded identification of 491 proteins that were enriched in the exosome surfaceome relative to TEE and cargo compartments (Fig. 4c). KEGG (Kyoto Encyclopedia of Genes and Genomes) and IPA (Ingenuity Pathway Analysis) pathway analyses revealed enrichment on the exosome surfaceome of proteins involved in phagocytosis/endocytosis and cell adhesion, as expected, but also proteins involved in antigen presentation and autoimmunity (Supplementary Fig. 5a). Interestingly, one of the most represented pathways was systemic lupus erythematosus (KEGG $p = 3.98 \times 10^{-8}$; IPA

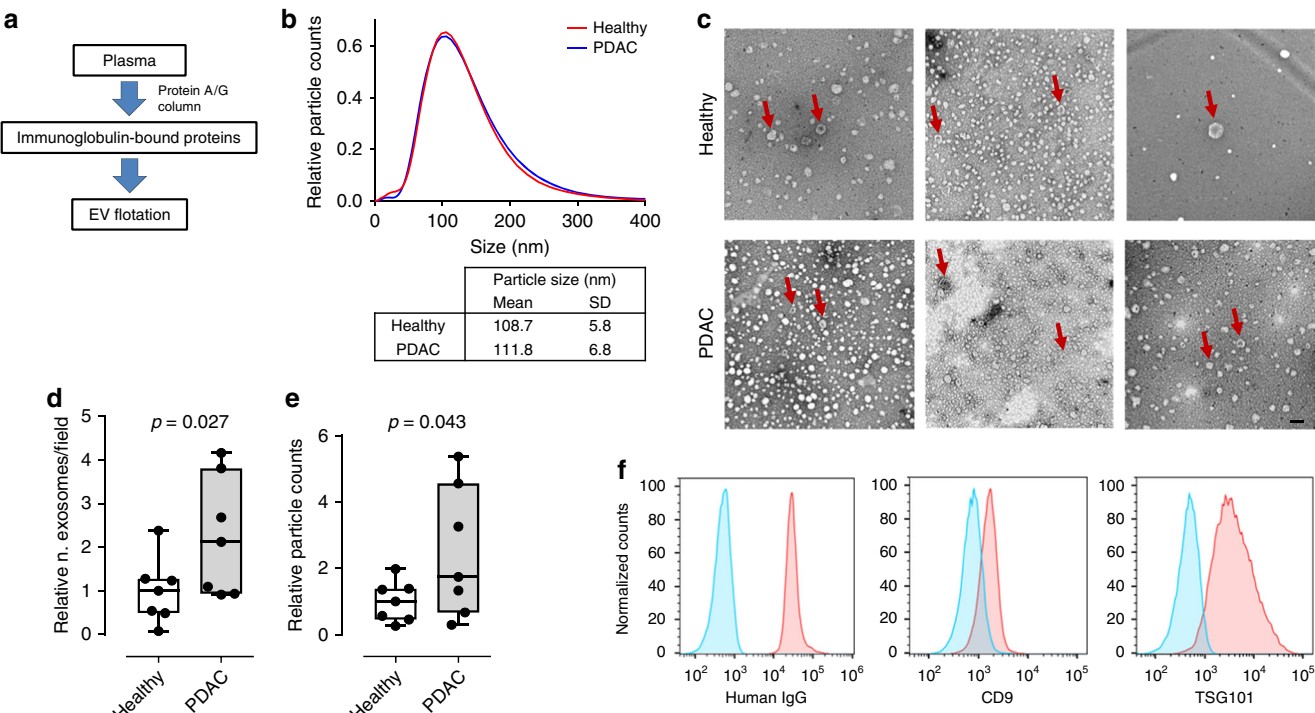

**Fig. 2** Characterization of exosomes bound to circulating immunoglobulins. **a** Schematic of the work-flow used for the isolation of exosomes present in the plasma Ig-bound fraction from PDAC patients and healthy controls (n = 7 per group). **b** Nanoparticle-tracking analysis of exosomes isolated from the plasma Ig-bound fraction indicating average size distribution of particles in PDAC and healthy control samples. Individual sample size distribution is shown in Supplementary Figure 2. **c** Representative TEM micrograph of exosomes isolated from the plasma Ig-bound fraction of PDAC patients and healthy controls. Arrows indicate vesicles with classical exosome size and morphology. Scale bars: 100 nm. **d** The relative number of exosomes per field (n = 3) analyzed by TEM. **e** The relative number of exosomes (30–200 nm size) in the Ig-bound fraction relative to the total number of exosomes in the neat plasma as quantified by nanoparticle-tracking (n = 3). Boxes in both graphs indicate 25th and 75th percentiles, and horizontal lines inside the boxes indicate median. Bars indicate 10th and 90th percentiles. Data were standardized based on the average of healthy controls. p-value was calculated by one-sided unpaired t-test. **f** Flow-cytometry analysis using anti-human IgG, CD63, and TSG101 of exosomes isolated from the plasma Ig-bound fraction and coupled to 0.4 μm-diameter beads. The graphs show representative analysis of a PDAC patient sample

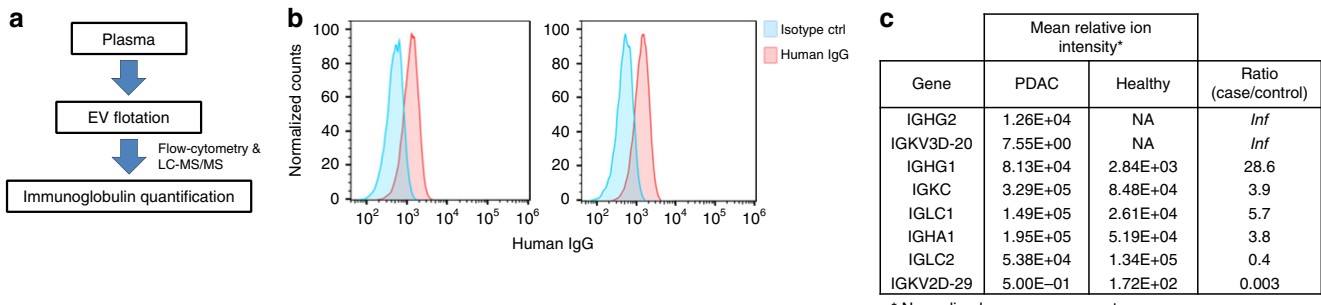

**Fig. 3** Immunoglobulins bound to circulating plasma exosomes. **a** Schematic of the work-flow used for the isolation and Ig quantification of plasma exosomes from plasma of PDAC patients and healthy controls. **b** Flow-cytometry analysis using anti-human IgG of exosomes isolated from the plasma of cancer patients coupled to 0.4 μm-diameter beads. The graphs show the representative analysis of two PDAC patient samples. **c** Table showing the relative spectral abundancy (ion intensity) in PDAC and healthy subject plasma exosomes of immunoglobulins. Numbers were normalized based on the number of exosomes (30–200 nm size) in each sample, as quantified by nanoparticle-tracking. Inf indicates proteins uniquely identified in cancer patient exosomes

$p = 5.89 \times 10^{-5}$). This was mostly due to the presence on the surface of PDAC exosomes of a number of nuclear proteins (i.e., histones and ribonucleoproteins) known to induce autoantibodies in autoimmune diseases (Supplementary Fig. 5b). We next investigated the presence within the exosome surfaceome of TAAs associated with autoantibodies in different tumor types annotated in the Cancer Immunome Database (ludwig-sun5.unil.ch/CancerImmunomeDB/). Enrichment analysis revealed a significant overlap between the PDAC exosome surfaceome and the CIDB ($p < 1.90 \times 10^{-8}$, hypergeometric test), suggesting a significant enrichment of TAAs within the exosome surfaceome (Supplementary Fig. 5c). Similar findings were obtained when the pathway analyses were restricted to proteins enriched in exosome surfaceome compared to the cell surface of PDAC cell lines (Fig. 4c and Supplementary Fig. 6).

**PDAC cell line and patient plasma exosome antigens concur.** We evaluated whether antigens identified by LC-MS/MS at higher

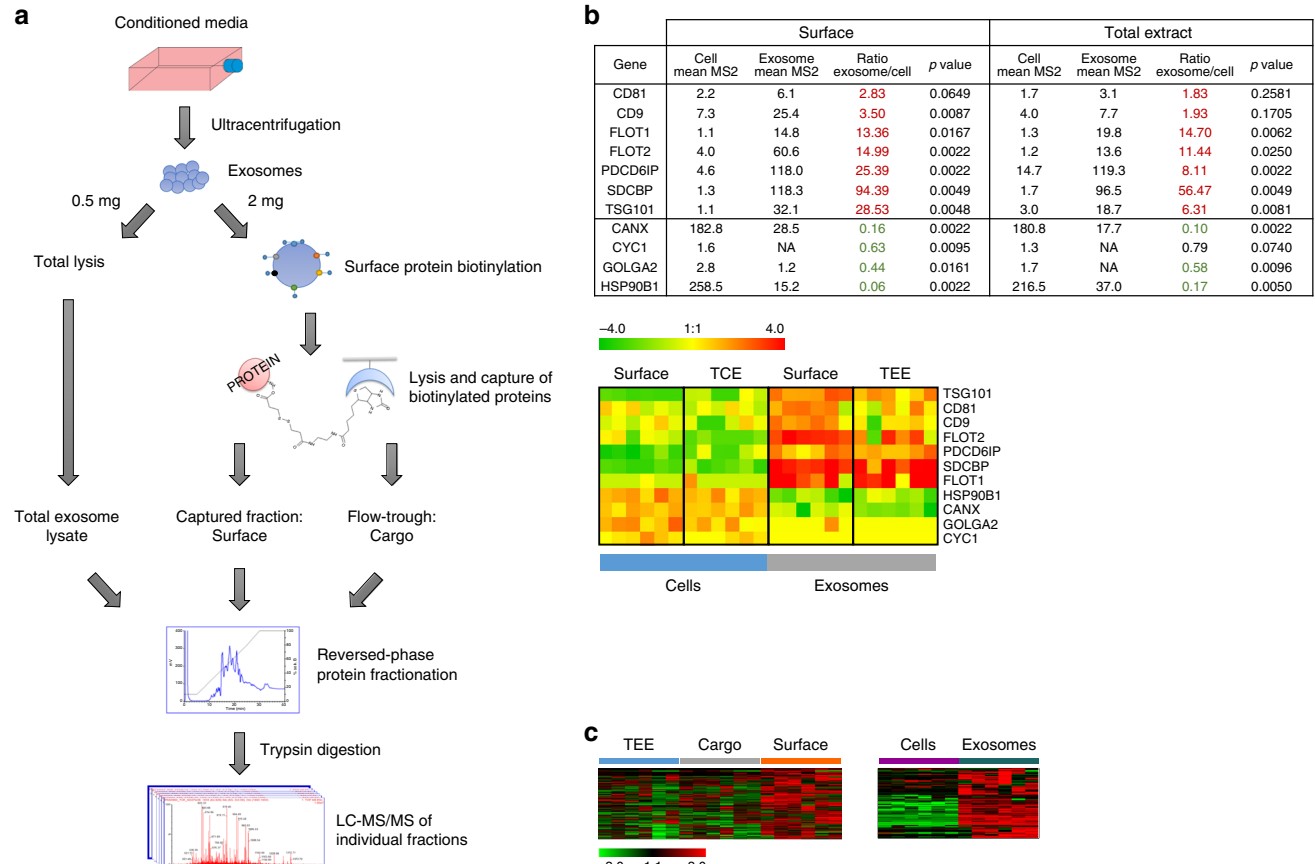

**Fig. 4** Proteomic profiling of PDAC cell line exosomes. **a** Schematic of the work-flow used for the proteomic analysis of PDAC cell line exosomes. Exosome surface proteins were obtained by labeling proteins on the outside of the vesicles with a biotinylated tag and then capturing these proteins by affinity chromatography. Proteins not bound to the column (flow through) were named as cargo proteins. In parallel with the total extract, these two sets of proteins were fractionated (by reversed-phase chromatography) and individual fractions subjected to tryptic digestion followed by LC-MS/MS, for protein identification. **b** Table and heatmap showing the average spectral abundance (MS2 counts) of exosome markers and endosomal proteins in total lysate and surface proteome of PDAC cells and exosomes. p-value was calculated by Mann–Whitney t-test. **c** Heatmaps depicting the cluster on proteins identified to be enriched (left panel) in the exosome surfaceome (orange bar) relative to total exosome extract (blue bar) and cargo compartments (gray bar) or on the exosome surface compared to cell surface (right panel). Heatmaps were generated through complete linkage hierarchical clustering of proteomics data (normalized MS2 counts); Pearson correlation was applied as the basis for distance measure. TCE total cell extract, TEE total exosome extract

levels in the Ig-bound fraction of PDAC patient plasma compared to controls were enriched in PDAC cell-derived exosomes. Thirty-four out of 92 Ig-bound proteins (37%; $p < 3.91 \times 10^{-4}$, hypergeometric test) were identified in PDAC cell and exosome surface or total extract proteome with at least five normalized MS2 count (Fig. 5a and Supplementary Table 5 and Data 2). Interestingly, this set of proteins exhibited significantly higher levels in exosomes compared to PDAC cells in both the surface fraction and total extracts ($p < 0.0001$, Fisher's exact test) and were significantly enriched in the exosome surfaceome compared to the TEE ($p = 0.014$, Fisher's exact test) (Fig. 5a, Supplementary Fig. 7 and Supplementary Table 5). We further assessed the antigenicity of these 34 exosomal proteins identified in the Ig-bound plasma fraction, as described below, by comparing data from PDAC cell line HLA-II immunopeptidome, PDAC plasma autoantibody reactivity against individual antigenic proteins using immunoblots and protein arrays, with the proteome of PDAC patient plasma circulating exosomes.

The presentation of TAAs on HLA class II molecules is necessary to induce T-lymphocyte activation and autoantibody production[17]. Tumor cells may express HLA-II complexes, which can promote T-cell activation after their transfer to antigen-presenting cells, which may occur in part through exosomes[18].

We assessed whether PDAC exosome proteins identified in the Ig-bound fraction (Fig. 5a) were presented on HLA-II molecules in PDAC cell lines. The same panel of PDAC cell lines from which exosomes were isolated was treated with 20 ng mL$^{-1}$ IFNγ for 24 h to enhance HLA-II expression. HLA-bound peptides were subsequently recovered by mild acid elution and analyzed by nano-LC-MS/MS. Following cell line HLA-II typing, we used the Immune Epitope Database ([www.iedb.org](http://www.iedb.org)) to assess which of the identified 12–34 amino acid peptide residues were predicted to be efficiently bound by HLA class II molecules. Peptides from 10 proteins present both in PDAC exosomes and the Ig-bound fraction were identified in one or more PDAC cell line with a predicted IC$_{50}$ < 500 nM (Supplementary Table 5 and Supplementary Fig. 8), providing further evidence for their antigenicity.

We next confirmed the reactivity of PDAC patient plasma against the 34 Ig-bound exosomal proteins. Protein lysates from the panel of six PDAC cell lines were pooled, separated using SDS-PAGE and subsequently immunoblotted with plasma pools ($n = 10$ samples per pool) from patients with early-stage PDAC, advanced PDAC, matched healthy subjects, and chronic pancreatitis controls (Supplementary Table 6, set #1). Bands specifically recognized by PDAC patient, but not control plasmas, were subjected to LC-MS/MS analysis (Supplementary Fig. 9a). A

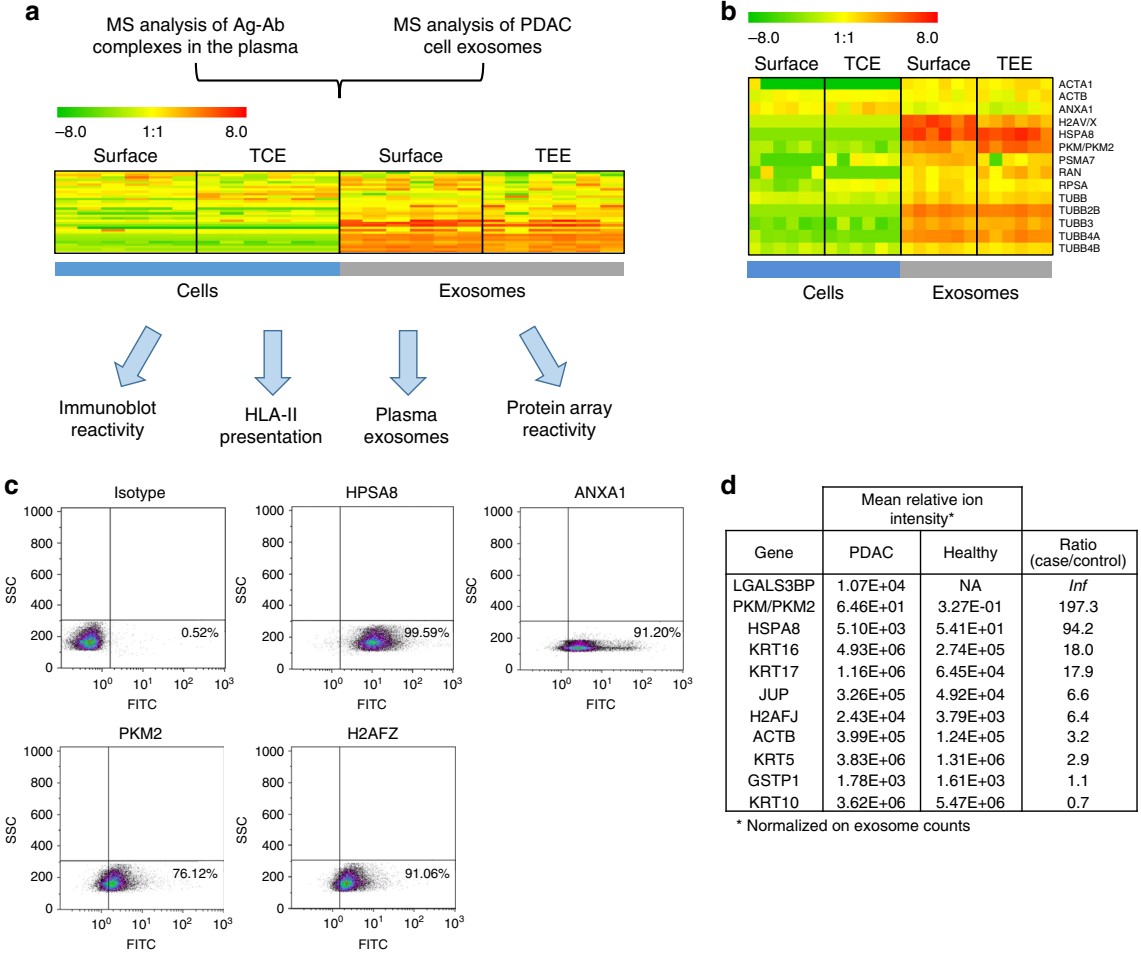

**Fig. 5** Exosomal antigens able to induce autoantibodies in PDAC patients. **a** Heatmap showing significant enrichment in the exosome compared to the cell proteome of antigens identified by MS analysis as bound to circulating Igs in the plasma of PDAC patients. Antigen average spectral abundance in each compartment is shown in Supplementary Table 5. **b** Heatmap showing the average spectral abundance (MS2 counts) in PDAC cell and exosome proteome of antigens identified in PDAC plasma Ig-bound fraction and immunoblot analysis of PDAC patient plasma reactivity. **c** Flow-cytometry analysis using anti-HSPA8, H2AFZ, PKM2, and ANXA1, or isotype control antibodies of PDAC cell line exosomes coupled to 0.4 μm beads. **d** Table showing the relative spectral abundancy (ion intensity) in PDAC and healthy subject plasma exosomes of Ig-bound exosomal proteins. Numbers were normalized based on the number of exosomes (30–200 nm size) in each sample, as quantified by nanoparticle-tracking. Inf indicates proteins uniquely identified in cancer patient exosomes. Ag antigen, Ab antibody, TCE total cell extract, TEE total exosome extract

total of 83 proteins with at least five MS2 counts were identified in the immunoreactive bands (Supplementary Table 7). Fourteen of these were also identified as Ig-bound exosome antigens and were enriched in PDAC cell line-derived exosomes compared to total cell proteomes ($p < 0.05$, Mann–Whitney $t$-test; Fig. 5b, Supplementary Fig. 8, and Supplementary Fig. 9b). Given our observation that autoantibody reactivity is preferentially directed at proteins contained in the exosome surfaceome, we evaluated the localization of a subset of these proteins: ANXA1, H2AF, PKM2, and HSPA8, previously reported as TAAs[19–22]. Immunogold TEM and flow-cytometry analysis confirmed occurrence of these four antigens on the surface of PDAC cell line-derived exosomes (Fig. 5c and Supplementary Fig. 10). Our comparative proteomic analyses provided compelling evidence that TAAs that induce an autoantibody response in PDAC patients are markedly enriched in tumor-derived exosome surfaceome.

We assessed the presence of TAAs in circulating exosomes isolated from the set of PDAC patients ($n = 6$) and matched healthy subject control plasmas ($n = 6$) described above (cohort #4; Supplementary Table 1 and Supplementary Fig. 3). LC-MS/MS analysis of PDAC patient and control plasma-derived exosomes yielded 1097 unique proteins identified in at least two

samples. Nineteen out of the 92 Ig-bound proteins (20.7%; $p < 3.14 \times 10^{-6}$, hypergeometric test), were present in plasma exosomes and, 11 were also identified in the PDAC cell line exosomes (Fig. 5d). Nine of these 11 Ig-bound proteins consisting of LGALSBP3, PKM/PKM2, HSPA8, KRT17, KRT16, ACTB, H2AF, KRT5, and JUP were elevated (>2.5-fold mean case-to-control increase in MS1 peptide ion intensity) in PDAC exosomes relative to matched controls (Fig. 5d and Supplementary Fig. 8). It is possible that the two other proteins may have been co-isolated in the Ig-bound fraction without being antigenic, or that their antigenicity in cancer might not uniquely depend on protein expression.

We next applied a targeted approach to assess autoantibody levels in PDAC patients against PKM2 and LGALSBP3, which were identified in the Ig-bound fraction and in circulating exosomes of PDAC patient plasma, by spotting the corresponding recombinant proteins on microarray slides for quantifying autoantibody reactivity. We tested a PDAC sample set resulting from the combination of three independent cohorts: set #1 consisted of stages IB to IIB PDAC cases ($n = 10$), healthy controls ($n = 10$), and chronic pancreatitis cases ($n = 10$); set #2 consisted of subjects who were subsequently diagnosed with

**Table 1 LGALS3BP and PKM2 autoantibody performance**

|  |  | LGALS3BP | PKM2 |
|---|---|---|---|
| PDAC vs. healthy | AUC | 0.647 | 0.696 |
|  | 95% CI | 0.554–0.739 | 0.604–0.787 |
|  | p-value | 0.001 | <0.0001 |
| PDAC vs. ch. pancreatitis | AUC | 0.638 | 0.663 |
|  | 95% CI | 0.545–0.732 | 0.564–0.762 |
|  | p-value | 0.002 | 0.001 |

AUC area under the curve, CI confidence interval, p-value Mann–Whitney t-test, ch. pancreatitis chronic pancreatitis

PDAC ($n = 13$) and matched healthy subject controls ($n = 13$) obtained from the Beta-Carotene and Retinol Efficacy Trial (CARET) pre-diagnostic cohort;[23] and set #3 consisted of early-stage (IA to IIA) PDAC cases ($n = 42$), healthy subjects ($n = 50$), and chronic pancreatitis cases ($n = 50$) (Supplementary Table 6). The samples were individually hybridized with recombinant protein microarrays for analysis. PDAC patients showed significantly higher levels of IgGs (AUC > 0.60; $p < 0.05$, Mann–Whitney t-test) against PKM2 and LGALSBP3 compared to matched healthy and chronic pancreatitis controls (Table 1 and Supplementary Fig. 8).

**Circulating autoantibodies bind to exosomes, attenuating CDC**. Given our findings of autoantibodies against antigens present on the surface of tumor-derived exosomes, we quantified reactivity against PDAC cell lines derived exosomes of Ig in plasmas from cases compared to controls. Exosomes isolated from the panel of six PDAC cell lines were pooled in equal numbers as quantified by nanoparticle-tracking analysis and used to test the Ig reactivity of PDAC patient and matched healthy control plasma samples. The PDAC cell line-derived exosomes were probed with Ig obtained by affinity purification from separate pools of 10 early-stage PDAC and 10 matched healthy subject control plasmas (Supplementary Table 6, set #1). Analysis based on labeling with protein A/G immunogold and TEM analysis, revealed significantly greater reactivity ($p = 0.007$, unpaired t-test) of patient compared to healthy subject against PDAC cell exosomes of different sizes, based on the average number of gold particles per exosome quantified in each TEM field (Fig. 6a).

To further validate our findings for Ig reactivity against exosomes in PDAC, we developed a quantitative bead-based assay for autoantibody reactivity against exosomes. We coupled exosomes to Luminex microspheres at an optimal concentration ($32 \mu g$ per $10^6$ beads), which was established using anti-CD63 and TSG101 antibodies after exosome titration (Fig. 6b, left panel). Pre-clinical PDAC samples from PDAC and samples from matched healthy subjects obtained from the CARET cohort (Supplementary Table 6, set #2) yielded a significantly higher level of autoantibody reactivity compared to controls (AUC = 0.787; 95% CI: 0.599–0.975; $p = 0.012$, Mann–Whitney t-test; Fig. 6b, right panel). We confirmed that the binding of antibodies to exosomes was independent of the Fc receptor (FcR) using an FcR-blocking reagent.

Autoantibodies can exert cytotoxic effects upon cancer cells after surface TAA ligation by promoting complement-dependent cytotoxicity (CDC) as well as antibody-dependent cell-mediated cytotoxicity. We next investigated whether cancer exosomes might provide a decoy function that modulates this immune response by competing for autoantibody binding, thus attenuating cytotoxicity against cancer cells. A pool of pre-diagnostic PDAC patient serum samples ($n = 13$) from the CARET study

showed an ability to effect CDC of tumor cells from the PANC-1 cell line. Exosomes isolated from the same cell line induced a significant dose-dependent inhibition of PDAC serum-mediated CDC ($p < 0.01$, unpaired t-test; Fig. 6c), providing evidence for such a decoy function. We next tested the decoy function of PDAC exosomes against CDC mediated by PDAC patient compared to healthy control sera from the pre-diagnostic CARET cohort. Live cell imaging cytotoxicity analysis using two PDAC cell lines: PANC-1 and Pa03C, established from metastatic tissue, revealed that exosomes isolated from the conditioned media of the respective cell lines were able to significantly ($p < 0.05$, unpaired t-test) attenuate the CDC mediated by PDAC patient serum samples, but not matched healthy controls (Fig. 6d and Supplementary Fig. 11). We observed no inhibitory effect of non-neoplastic exosomes isolated from an immortalized PDAC-associated fibroblast line (CAF19) on the CDC mediated against Pa03C cells by PDAC sera and matched healthy controls from the CARET cohort (Fig. 6d).

**Discussion**

In this study, we have applied an unbiased approach, based on high-throughput proteomic technologies, to perform orthogonal analyses of plasma samples from a number of independent PDAC cohorts, as well as cancer cell lines in order to investigate the underlying mechanisms responsible for the generation of auto-antibodies against intracellular TAAs. Here we have demonstrated, through specific surfaceome profiling, that tumor-derived exosome surfaces are enriched with TAAs able to induce auto-antibodies and provided evidence of a decoy function of exosomes against tumor cell complement-mediated cytotoxicity.

We identified circulating plasma exosomes bound to immunoglobulins in the plasma of PDAC patients. Exosomes were also detected, although at lower levels, in the circulating immunoglobulin-bound fraction of healthy subjects, likely related to inflammatory conditions, allergies or autoimmune diseases. Consistently, increased levels of immunoglobulins were quantified in the exosome-enriched fraction of PDAC patient compared to matched healthy control plasmas after density floatation of exosomes. The proteomic profiling of exosomes isolated from a panel of PDAC cell lines indeed revealed enrichment in the exosome surfaceome of numerous antigens able to induce a humoral response in tumors as well as in systemic lupus erythematosus, suggesting that exosomes might be involved in triggering an autoantibody response not only in cancer but also in autoimmune diseases. In line with our observation, two recent studies have identified EVs with a key role for production of anti-perlecan antibodies in organ transplant recipients[22] and of anti-citrullinated peptide antibodies in rheumatoid arthritis[24]. The high expression on the exosome surface of intracellular antigens that are not usually membrane-associated may arise through exosome-specific biogenesis and protein-sorting mechanisms, or through the binding of exosome-enriched adhesion and proteoglycan molecules.

Our findings suggest that exosomes might expose B-cell epitopes. It has indeed been demonstrated in a fibrosarcoma tumor system expressing a model antigen (OVA), both in vitro and in vivo, that only the exosome surface-bound form of OVA was able to elicit a strong antigen-specific CD8[+] T-cell response, helper CD4[+] T-cell activation, antigen-specific antibodies, and a decrease in the percentage of immunosuppressive regulatory T cells in tumors[25]. A robust T-cell response was not observed when antigens were expressed in a secreted soluble form or as a cell membrane-bound protein. Consistent with these findings, Qazy et al.[26] have demonstrated that, although potent antigen-specific CD8[+] T-cell activation was induced in vitro by both

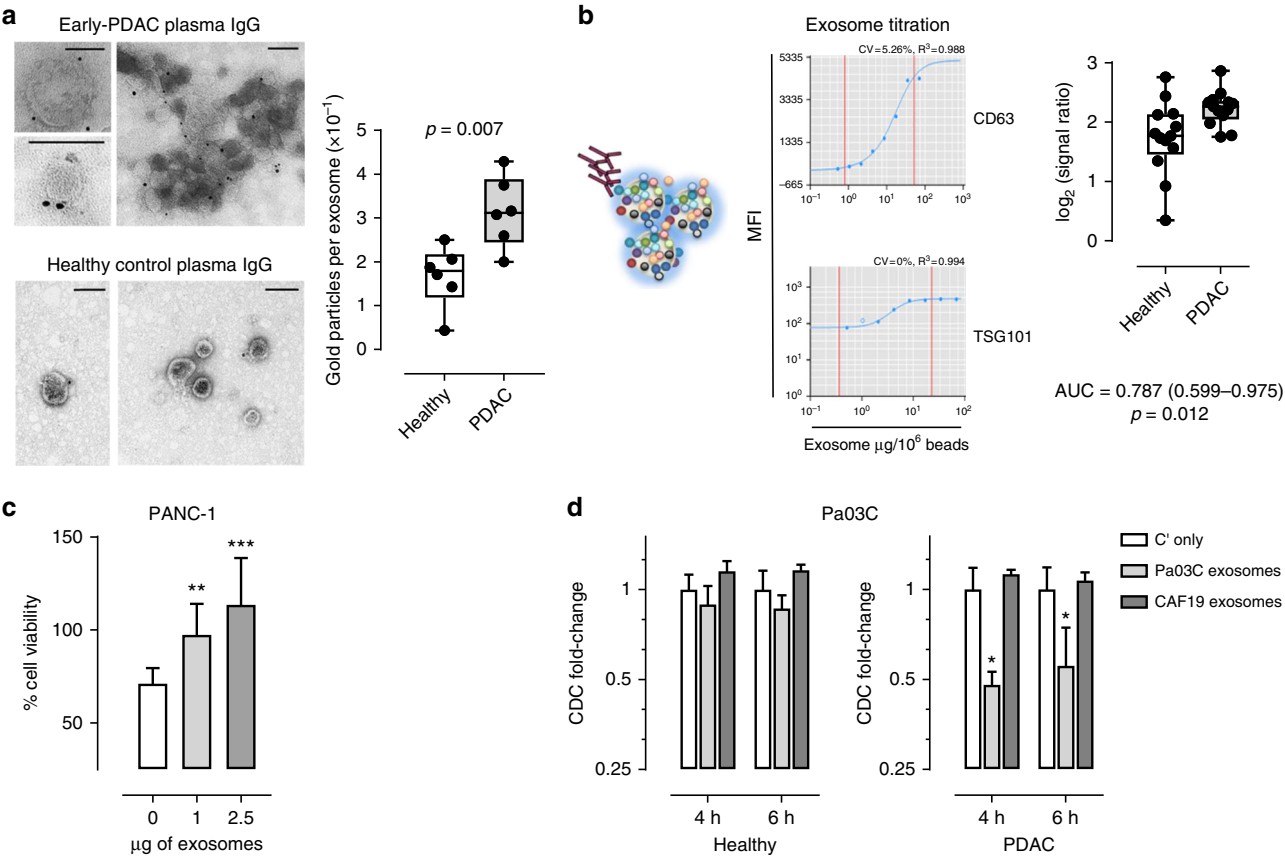

**Fig. 6** Binding of PDAC patient autoantibodies to PDAC cell exosomes. **a** Representative TEM micrograph of the immunogold labeling of exosomes isolated from PDAC cell lines using IgGs affinity-purified from early-stage PDAC (pool, $n = 10$) and matched healthy subject control (pool, $n = 10$) plasmas. Black dots indicate gold immunolabeling. Scale bars: 100 nm. Graph represents the average number of gold particles per exosome in each field analyzed by TEM ($n = 6$). Boxes indicate 25th and 75th percentiles; horizontal lines indicate median. Bars are 10th and 90th percentiles. $p$-value was calculated by unpaired $t$-test. **b** Luminex assay of PDAC patient autoantibody reactivity against PDAC cell-derived exosomes. An equal number of exosomes isolated from the panel of 6 PDAC cell lines were combined and coupled to Luminex microspheres at optimal concentration established using anti-CD63 and TSG101 antibodies after exosome titration (left panels). Graph showing autoantibody reactivity of PDAC and matched healthy subject serum samples from the CARET cohort ($n = 13$ per group) against PDAC cell line-derived exosomes coupled to Luminex beads (right panel). Boxes indicate 25th and 75th percentiles; horizontal lines inside the boxes indicate median. Bars are 10th and 90th percentiles. $p$-value calculated by Mann–Whitney $t$-test ($n = 3$). **c** Complement-dependent cytotoxicity of PANC-1 cells, in presence or absence of increasing amount of PANC-1 exosomes, mediated by pre-diagnostic PDAC sera from the CARET study (pool, $n = 13$). Graphs illustrate the mean result of three independent experiments ± SD. **d** Complement-dependent cytotoxicity of Pa03C cells, in presence of Pa03C or CAF19 exosomes, mediated by pre-diagnostic PDAC sera or matched healthy controls from the CARET study (pool, $n = 13$). Experiments were performed by live cell imaging cytotoxicity analysis. Data are expressed as fold change in the number of green (dead) cells relative to the respective controls (sera plus complement in absence of exosomes; C′ only) at two different time points after complement incubation. Graphs illustrate the mean result ± SD of triplicates from a representative experiment of three replicates. $p$-values calculated by unpaired $t$-test: $*p < 0.05$, $**p < 0.01$ and $***p < 0.001$. AUC area under the curve

peptide- and whole-antigenic protein-loaded DC exosomes, only the latter, containing both T- and B-cell epitopes, induced specific cytotoxic T-lymphocyte responses in vivo. A second study from the same group reported that exosome-mediated immune activation needs the assistance of B cells, in particular marginal zone B cells (MZBs), for generating in vivo antigen-specific T-cell responses[27]. MZBs are closely associated with the marginal sinus of the spleen and shuttle blood-borne antigens into the splenic B-cell zone, where they deposit antigens on follicular DCs, leading to activation of both B and CD4+ T cells[28]. Given that splenic MZBs catch and transport antigens in a complement-dependent manner, the authors hypothesized that the transport of exosomes by MZBs might also be mediated by complement. However, based on our data this mechanism might instead be antigen-dependent. Taken together, these results suggests that B-cell native epitopes on exosomes might be more efficiently presented and protected from proteolysis compared with the soluble form of

the protein, making them more accessible to B cells, and leading to a coordinated CD4+ and CD8+ T-cell response and the secretion of autoantibodies.

Nevertheless, trials to evaluate the use of tumor exosome-loaded DCs for immunostimulation have shown limited clinical antitumor efficacy, and there is increasing data that tumor-derived EVs seem indeed to have a predominantly immunosuppressive role. There is substantial evidence that tumor exosomes inhibit antigen-specific and non-specific antitumor responses by suppressing activation of T- and NK-cells, promoting the generation of myeloid-derived suppressor cells and enhancing T-regulatory cell function[13]. We propose that tumor exosomes might promote B-cell responses that favor immune escape. For instance, exosomes could promote the expansion of immunosuppressive B-cell populations. Multiple studies have demonstrated that tumor-infiltrating lymphocytic B cells enhanced tumor progression through a subset of B cells with

immunosuppressive functions[29,30]. Moreover, we observed that PDAC exosomes by exposing several TAAs exert a broad decoy-like function by binding circulating autoantibodies present in PDAC patient serum, thereby inhibiting complement-dependent cytotoxicity and potentially antibody-dependent cell-mediated cytotoxicity. In line with our findings, it has been demonstrated that tumor-derived exosomes promote resistance to cell surface directed anti-CD20 and anti-Her2 therapeutic antibodies by mediating their sequestration and thus impairing the specifically induced autoantibody-based cytotoxicity directed at cancer cells[31–33]. These observations provide novel insights into possible immune escape mechanisms mediated by tumor-derived exosomes, which deserve further investigation and might explain the limited success of exosome-based cancer immunotherapies. Cancer exosomes could then be targeted to limit immune suppression, potentially increasing efficacy of current therapies. Approaches to target tumor exosomes might also be based on small-molecule inhibition of exosome biogenesis or secretion[34]. An alternative approach would be to selectively deplete exosomes from the circulation using cancer specific exosome markers to affinity-capture tumor secreted vesicles[35–37].

Through our integrated analysis of PDAC cell line exosome proteome, immunoglobulin-bound proteins and cancer patient autoantibody reactivity, we identified a number of exosome surface-associated TAAs. Many of these are enzymes or cytoskeletal proteins (ANXA1, KRT10, KRT16, TUBB, ACTB, ACTA1, PKM2, and HSPA8) that have been previously reported as able to induce autoantibodies with diagnostic potential across different tumor types, including PDAC, breast cancer, and neuroblastoma[19–21,38]. Several PDAC cell exosome-associated antigens present in the immunoglobulin-bound fraction (LGALSBP3, PKM2, HSPA8, KRT17, KRT16, ACTB, H2AF, KRT5, and JUP) also showed enrichment in PDAC patient plasma EVs relative to healthy subjects, providing evidence of potential exosome-associated PDAC biomarkers.

We demonstrated that PDAC patient autoantibodies show significantly higher reactivity against whole tumor exosomes, compared to matched controls. We then assessed whether autoantibodies against exosome surface TAAs might have diagnostic value in PDAC, using a protein microarray-based approach. By combining three independent plasma sample cohorts, we identified significantly higher levels of autoantibodies against two exosome surface proteins (PKM2 and LGALSBP3) in early-stage PDAC patients compared to matched healthy subject and chronic pancreatitis controls. These results suggest that circulating autoantibodies against TAAs present on exosome offer diagnostic utility and merit further assessment.

In conclusion, our work has provided novel insights into the role of tumor exosomes in modulating the immune response, with important implications for PDAC therapy and early detection that deserve further exploration in this and other tumor types.

## Methods

**Blood samples**. All human blood samples were obtained under respective MD Anderson, Evanston Hospital, University of Utah and CARET (Beta-Carotene and Retinol Efficacy Trial) site Institutional Review Board approval and informed consent of all participants was obtained. For MS analysis of Ig-bound proteins, we applied two PDAC sample cohorts (Fig. 1b): (i) a cohort from the University of Texas MD Anderson Cancer Center (cohort #1, Supplementary Table 1), consisting of potentially resectable ($n = 18$) and locally advanced ($n = 10$) PDAC patients and benign pancreatic cyst patients ($n = 12$); and (ii) a cohort from the University of Utah (cohort #2, Supplementary Table 1), consisting of stage I ($n = 12$) and stage II ($n = 20$) PDAC patients, healthy subjects ($n = 32$) and chronic pancreatitis patients ($n = 32$). A sample set from the University of Texas MD Anderson Cancer Center consisting of advanced PDAC patients ($n = 12$) and healthy subjects ($n = 12$) was applied for the analyses of exosomes isolated from the Ig-bound fraction (cohort #3, Supplementary Table 1). Another cohort (cohort #4, Supplementary Table 1)

from the University of Texas MD Anderson Cancer Center consisting of advanced PDAC patients ($n = 6$) and healthy subjects ($n = 6$) was applied for the proteomic profiling of plasma exosomes. Three sets of PDAC plasma samples were applied for autoantibody analysis using recombinant protein arrays and additional assays described in the study (Supplementary Table 6): (i) set #1 from Evanston Hospital consisting of early-stage PDAC cases ($n = 10$), healthy subjects ($n = 10$) and chronic pancreatitis patients ($n = 10$); (ii) set #2 consisting of a pre-diagnostic cohort from subjects that were subsequently diagnosed with PDAC ($n = 13$) and matched healthy subjects ($n = 13$), as part of the Carotene and Retinol Efficacy Trial (CARET);[23] and (iii) set #3 from the University of Utah consisting of early-stage PDAC cases ($n = 42$), healthy subjects ($n = 50$) and chronic pancreatitis patients ($n = 50$). All samples were collected at diagnosis when patients were treatment naive. In order to control for possible confounding factors, in all the cohorts PDAC cases and controls were matched for age, gender and type 2 diabetes status.

**Cell culture**. Eleven PDAC cell lines (SU.86.86, PANC-1, CFPAC-1, MIA PaCa-2, Panc 03.27, HPAF-II, BxPC-3, SW 1990, Hs 766 T, AsPC-1, and Capan-2) were purchased from ATCC. Pa03C is a lab-established (KW Kinzler Lab) PDAC cell line from liver metastatic tissue[39]. CAF19 is an immortalized cancer-associated fibroblast line derived (M Goggins Lab) from a patient pancreatic adenocarcinoma[40]. The identity of each cell line was confirmed by DNA fingerprinting via short tandem repeats at the time of mRNA and total protein lysate preparation using the PowerPlex 1.2 kit (Promega). Fingerprinting results were compared with reference fingerprints maintained by the primary source of the cell line. All cell lines were routinely tested and confirmed negative for mycoplasma contamination using a MycoAlert mycoplasma detection kit (Lonza). For MS analysis, cells were grown in RPMI or DMEM media for SILAC (Pierce) containing 10% dialyzed fetal bovine serum (FBS) (Life Technologies) and 0.1 mg mL$^{-1}$ $^{13}$C–lysine (Cambridge Isotope Laboratories, Inc.) for seven passages.

**Isolation and purification of exosomes**. Cell line exosome were purified by differential centrifugation as previously described[15]. Briefly, supernatant from PDAC cell lines (CFPAC-1, HPAF-II, SU.86.86, Panc 03.27, MIA PaCa-2, and PANC-1) cultured for 48 h in serum-free media were subjected to sequential centrifugation steps of $800 \times g$ and $2000 \times g$. The resulting supernatant was filtered using 0.2 µm filter and concentrated 50 times using a Vivaflow 200 R crossflow unit (Sartorius) with a 100,000 KDa cutoff filter. A pellet was recovered after 2 h ultracentrifugation at $100,000 \times g$ in a 45Ti fixed angle rotor (Beckman Coulter). Supernatant was removed and PBS added to the pellet for an overnight washing step. The resultant exosome pellet was resuspended in PBS and harvested for downstream analyses. For the isolation of plasma or Ig-bound fraction exosomes we applied a density gradient flotation approach. Microvesicles were depleted by centrifugation at $2000 \times g$ for 20 min followed by $16,500 \times g$ for 30 min; the resulting supernatant was additionally filtered through a pre-wetted 0.22 µm vacuum filter (Steriflip SCGP00525, Millipore). Microvesicle-depleted plasma was mixed with OptiPrep iodixanol solution (Sigma-Aldrich) to a final density of $1.17$ g mL$^{-1}$. This was loaded into the bottom of a polycarbonate ultracentrifuge tube (Seton Scientific) and overlaid with ~3 mL of $1.14$ g mL$^{-1}$ iodixanol/PBS solution to form a single-step density fractionation gradient. Ultracentrifugation was performed at $100,000 \times g$ for 16 h at 8 °C. Vesicles were collected in a single fraction from the top of the tube, proceeding downward to recover 0.6 mL of overlaid gradient volume. Density of each harvested fraction was assessed against a standard curve based on sample absorbance at 250 nm using a NanoDrop microvolume spectrophotometer (Thermo Fisher Scientific).

**Particle-size distribution and quantification**. Samples of EVs were quantified via Brownian diffusion size analyses using ZetaView Nanoparticle-tracking analysis (NTA) instrumentation (Particle Metrix). Sample aliquots were diluted $10^2$–$10^6$ fold to achieve optimal concentration for analysis; 1.0 mL of diluted sample was used for each analysis. Light scattering of individual particles in solution was digitally recorded, particle trajectory and displacement were automatically analyzed by image analysis tracking software, and the particle-size distribution was determined from the observed Brownian motion of individual particles according to the Stokes–Einstein relationship.

**Flow-cytometry analysis**. Exosome flow-cytometry analysis was performed as previously described[15]. Briefly, exosomes preparations (15 µg) were incubated with 10 µL of 4-µm-diameter aldehyde/sulfate latex beads (Life Technologies) for 15 min at RT, then resuspended in 1 mL PBS and incubated on a test tube rotator wheel overnight at 4 °C. Free binding sites on beads were saturated with 110 µl of 1 M glycine at RT for 30 min. Exosomes-coated beads were resuspended into 400 µL PBS containing 2% BSA and a 25 µL aliquot was incubated with the following antibodies: CD63 (sc-5275, Santa Cruz), CD9 (#655433, BD Bioscience), TSG101 (ab83, Abcam), ANXA1 (AF3770, R&D Biosystems), H2AFZ (sc-67218, Santa Cruz), HSPA8 (NB100-41377, Novus), PKM2 (D78A4, Cell Signaling), and anti-human IgG (#409309, BioLegend), or an isotype-matched negative control antibody for 30 min at 4 °C followed, when needed, by incubation with FITC-conjugated secondary antibody (BD Bioscience). One microgram of primary and

secondary antibody was used for all stainings. Flow-cytometry data were acquired on a Gallios flow cytometer (Beckman Coulter) and analyzed using FlowJo.

**Immunogold labeling and electron microscopy**. Exosome aliquots were fixed in 2% paraformaldehyde; 5 μL of exosome suspension was placed onto a 400 mesh carbon/formvar coated grids, previously treated with poly-L-lysine, and allowed to absorb to the formvar for 1 h. For immunogold staining the grids were placed into a blocking buffer (PBS, 2% BSA) for 30 min. Without rinsing, the grids were immediately placed into the primary antibody or an isotype-matched negative control antibody at the appropriate dilution overnight at 4 °C. The following primary antibodies were applied: CD63 (sc-5275, Santa Cruz), ANXA1 (AF3770, R&D Biosystems), H2AFZ (sc-67218, Santa Cruz), HSPA8 (NB100-41377, Novus), PKM2 (D78A4, Cell Signaling) at a 1:5 dilution, and IgG purified from plasma using Pierce Protein A/G Agarose (Thermo Fisher Scientific) at a 1:50 dilution. The next day all of the grids were rinsed with PBS then floated on drops of Protein A and Protein G attached with 6 nm and 10 nm gold particles, respectively (1:10 dilution; AURION), for 2 h at RT. Grids were rinsed with PBS and were placed in 2.5% glutaraldehyde in 0.1 M Phosphate buffer for 15 min. Samples were then negatively stained with filtered aqueous 1% uranyl acetate for 1 min. Excess uranyl acetate was blotted from the grid edge with Whatman No. 1 filter paper, and the grids were air-dried. Samples were then examined in a JEM 1010 transmission electron microscope (JEOL USA Inc.) at an accelerating voltage of 80 Kv. Digital images were obtained using the AMT Imaging System (Advanced Microscopy Techniques Corp.). ImageJ software have been used to count exosomes in each field.

**Western blot analysis**. To obtain protein lysate cells were resuspended in RIPA buffer and exosomes in 8 M Urea + 2.5% SDS. The lysates were then separated by SDS-PAGE and subjected to western blotting. Western blot analysis was performed according to standard procedures using polyvinylidene difluoride membranes and an enhanced chemiluminescence system (GE Healthcare). The following antibodies were used: CD63 (EXOAB-CD63A-1, System Biosciences, 1:1000), CD9 (EXOAB-CD9A-1, System Biosciences, 1:500), Alix (#2171, Cell Signaling, 1:500), and TSG101 (EXOAB-TSG101-1, System Biosciences, 1:500). To test plasma samples reactivity against exosome lysate, blotted membranes were incubated with plasmas at a 1:100 dilution overnight at 4 °C and then with horseradish peroxidase (HRP)-conjugated rabbit anti-human IgG (1 h, 1:4000; GE Healthcare).

**Isolation and fractionation of Ig-bound proteins in plasma**. Ig-bound proteins from a total of 100 μL (for mass spectrometry analysis) or 2 mL (for exosome isolation) of plasma were extracted using NAb™ Protein A/G spin columns (Thermo Fisher Scientific) according to manufacturer's instructions. To process 100 μL of plasma, columns were equilibrated twice with 400 μL binding buffer (Phosphate Buffered Saline, pH 7.2) and then incubated for 10 min at RT with plasma samples diluted 1:2 in PBS, pH 7.2. Columns were washed three times with 400 μL of PBS, pH 7.2. Ig-bound proteins were eluted twice with 400 μL of 0.1 M glycine, pH 3; the flow-through was collected and then neutralized with 40 μL of PBS, pH 9. After each step columns were centrifuged for 1 min at 5000 × g. To test the non-specific binding of exosomes to the Protein A/G spin columns, an additional low pH wash with 400 μL of PBS, pH 5, was performed before Ig-bound elution as presented in Supplementary Fig. 1.

For mass spectrometry analysis, the collected proteins were treated with 25 mM TCEP for Cys reduction and subsequently alkylated with acrylamide. The samples were next fractionated at the protein level by reverse-phase chromatography followed by desalting for 5 min with 95% mobile phase A (0.1% TFA in 95% H₂O). Proteins were eluted from the column and collected into 12 fractions, with a gradient elution that included an increase from 5 to 70% mobile phase B (0.1% TFA in 95% acetonitrile) over 25 min, 70 to 95% mobile phase B for 3 min, a wash step to hold at 95% mobile phase B for 2 min, followed by a re-equilibration step at 95% mobile phase A for 5 min.

**Cell line exosome fractionation**. Exosome surface, cargo and total extract proteins were isolated from the same sample. Total exosome extract proteins was obtained by sonication of exosome pellet in 1 mL of 4 M urea, 3% isopropanol, 20 mM Tris containing the detergent octyl-glucoside (OG) (2% wt/vol) and protease inhibitors (complete protease inhibitor cocktail, Roche Diagnostics) followed by centrifugation at 20,000 × g at 4 °C for 30 min. To isolate exosome surface from cargo proteins, exosome pellet was biotinylated with 5 mL of 1 mg mL⁻¹ of Sulfo-NHS-SS-BIOTIN (Pierce) in PBS for 30 min at 4 °C. The residual biotinylation reagent was quenched with 10 mL of 100 mM lysine in cold PBS for 15 min at 4 °C. Biotinylated exosomes were recovered through ON ultracentrifugation at 100,000 g. Biotinylated exosomes were then sonicated in 2 mL of 4 M urea, 3% isopropanol, 20 mM Tris, 2% OG, and protease inhibitors followed by centrifugation at 20,000 × g at 4 °C for 30 min. Biotinylated proteins were isolated by affinity chromatography using 2 mL of UltraLink Immobilized Neutravidin (Pierce) according to manufacturer instruction. Proteins bound to the column were recovered by reduction of the biotinylation reagent with 1 mL of a solution containing 65 μmol of DTT and 2% (wt/vol) OG detergent overnight at 4 °C and referred to as exosome surface proteins. Proteins not bound to the column (flow through) were also collected and named cargo proteins.

Exosome surface, cargo and total exosome lysate proteins were fractionated by reversed-phase chromatography, using the same amount of proteins across different samples for a given exosome compartment. All three extracts were reduced by DTT and alkylated with acrylamide prior to chromatography. Separation were performed in an off-line 1100 series HPLC system (Shimadzu) with reversed-phase column (4.6 mm ID × 150 mm length, Column Technology Inc.) at 2.7 mL min⁻¹ using a linear gradient of 10 to 80% of organic solvent over 30 min run. Solvent system used was: aqueous solvent—5% acetonitrile/95% water/0.1% of trifluoroacetic acid; organic solvent—75% acetonitrile/15% isopropanol/10% water/0.095% tri-fluoroacetic acid. Fractions were collected at a rate of 3 fractions per min.

**Cell line subcellular fractionation**. Cell surface and TCE proteins were extracted from the same batch of cells. Total cell extract proteins were obtained by sonication of ~2 × 10⁷ cells in 1 mL of PBS containing the detergent octyl-glucoside (OG) (1% wt/vol) and protease inhibitors (complete protease inhibitor cocktail, Roche Diagnostics) followed by centrifugation at 20,000 × g at 4 °C for 30 min. Secreted proteins were obtained from conditioned media (from ~2 × 10⁷ cells) centrifuged at 5000 × g and filtrated through a 0.22 μm filter to remove cells and debris. Secreted proteins were then concentrated to 1 mL using a 3 kDa molecular weight cutoff centrifugal filter (Millipore). To isolate cell surface proteins, ~2 × 10⁸ cells were rinsed 3 times with 10 mL of cold PBS and biotinylated for 10 min in the culture plates with 10 mL of 0.25 mg mL⁻¹ of Sulfo-NHS-SS-BIOTIN (Pierce) in PBS at room temperature. The residual biotinylation reagent was quenched with 10 mL of 10 mM lysine in cold RPMI media for 5 min at room temperature. Cell surface proteins were obtained by sonication of the biotinylated cells in 2% NP-40 detergent in PBS (vol/vol) followed by centrifugation at 20,000 × g at 4 °C for 30 min. Biotinylated proteins were isolated by affinity chromatography using 1 mL of UltraLink Immobilized Neutravidin (Pierce) according to manufacturer instruction. Proteins bound to the column were recovered by reduction of the biotinylation reagent with 1 mL of a solution containing 65 μmol of DTT and 1% (wt/vol) OG detergent overnight at 4 °C. Cell surface, conditioned media and total cell lysate proteins were fractionated by reversed-phase chromatography as described above. Each fraction from the reverse-phase chromatography was in-solution digested overnight at 37 °C with 400 ng of trypsin. The resulting trypsi-nized fractions were pooled into 4–10 pools based on chromatographic features. Pools were individually analyzed by LC-MS/MS.

**SDS-PAGE analysis**. Coomassie G-stained bands were excised from SDS-PAGE gels for individual profiling. Gel fractions were minced into 1 mm³ cubes; destained for 2 h in a 1:1 mix of 100 mM NH₄HCO₃/acetonitrile; alkylated for 30 min with 0.154% DTT in 100 mM NH₄HCO₃; additionally alkylated for 30 min with 0.271% acrylamide in 100 mM NH₄HCO₃; followed by washing for 30 min with 5% acetic acid in methanol. Following aspiration of wash buffer, the gel pieces were washed additionally in 100 mM NH₄HCO₃; followed by aspiration and final wash with acetonitrile. The gel pieces were then dried 45 min using a SpeedVac vacuum concentrator and subjected to overnight tryptic digestion. Digested peptides were extracted from the gel pieces using 20 μL of a 1:1 mix of 0.1% TFA in H₂O/acetonitrile for 15 min; followed by spin and supernatant harvest and a second extraction using 20 μL of a 1:2 mix 0.1% TFA in H₂O/acetonitrile for 15 min. The pooled supernatants were vacuum concentrated to dryness and stored at −80 °C for subsequent LC-MS/MS analyses.

**HLA-bound peptide extraction**. HLA-bound peptides were eluted from 5 × 10⁸ cells of CFPAC-1, HPAF-II, SU.86.86, Panc 03.27, MIA PaCa-2, PANC-1 cell lines as previously described[47]. Cell lines were incubated with or without 20 ng mL⁻¹ IFNγ for 24 h before elution of HLA-bound peptides. In brief, 4 mL of citrate-phosphate buffer at pH 3.3 (0.131 M citric acid; 0.066 M Na2HPO4; NaCl 150 mM) containing a protease inhibitor mixture and phosphatase inhibitors was added to each dish. Cells were scraped gently from the plate and pooled together and resuspended by gentle pipetting for 1 min to denature HLA-peptide complexes. Cell suspensions were then pelleted, and the resulting supernatant was isolated. Peptides were then passed through ultrafiltration devices (3-kDa cutoff, Amicon Ultra; Millipore) to isolate peptides from β2m proteins. Peptides obtained after acid elution were desalted and separated using an off-line 1100 series HPLC system (Shimadzu) with reversed-phase column (4.6-mm internal diameter × 150-mm length; Column Technology). Peptides were fractionated with a 41-min elution program at a flow rate of 2.1 mL per min: 0–2% B (0–5 min), 2–35% B (5–30 min), 35–50% B (30–33 min), 50–95% B (33–35 min), 95% B (35–37 min), 95–2% B (37–37.5 min), 2% B (37.5–41 min). Mobile phase A: 5% acetonitrile, 95% H₂O, 0.1% trifluoroacetic acid (TFA); Mobile phase B: 5% H₂O, 95% acetonitrile, 0.1% TFA. The fractionated peptides were collected in 96 consecutive fractions, lyophilized and subsequently resuspended in 50 μL of 1%.

TFA after pooling into 12 pools of consecutively eluted fractions and lyophilized.

**Mass spectrometry analysis**. For cell line exosomes, plasma exosomes and Ig-bound proteins, protein digestion and identification by LC-MS/MS was performed using our established protocol[41]. Briefly, NanoAcquity UPLC system coupled in-line with WATERS SYNAPT G2-Si mass spectrometer was used for the

separation of pooled digested protein fractions. The system was equipped with a Waters Symmetry C18 nanoAcquity trap-column (180 μm × 20 mm, 5 μm) and a Waters HSS-T3 C18 nanoAcquity analytical column (75 μm × 150 mm, 1.8 μm). The column oven temperature was set at 50 °C, and the temperature of the tray compartment in the auto-sampler was set at 6 °C. LC-HDMSE data were acquired in resolution mode with SYNAPT G2-Si using Waters Masslynx (version 4.1, SCN 851). The capillary voltage was set to 2.80 kV, sampling cone voltage to 30 V, source offset to 30 V and source temperature to 100 °C. Mobility utilized high-purity N2 as the drift gas in the IMS TriWave cell. Pressures in the helium cell, Trap cell, IMS TriWave cell and Transfer cell were 4.50 mbar, $2.47 \times 10^{-2}$, 2.90, and $2.53 \times 10^{-3}$ mbar, respectively. IMS wave velocity was 600 m s$^{-1}$, helium cell DC 50 V, Trap DC bias 45 V, IMS TriWave DC bias V and IMS wave delay 1000 μs. The mass spectrometer was operated in V-mode with a typical resolving power of at least 20,000. All analyses were performed using positive mode ESI using a NanoLockSpray source. The lock mass channel was sampled every 60 s. The mass spectrometer was calibrated with a [Glu1] fibrinopeptide solution (300 fmol μL$^{-1}$) delivered through the reference sprayer of the NanoLockSpray source. Accurate mass LC-HDMSE data were collected in an alternating, low energy (MS) and high energy (MSE) mode of acquisition with mass scan range from m/z 50 to 1800. The spectral acquisition time in each mode was 1.0 s with a 0.1-s inter-scan delay. In low energy HDMS mode, data were collected at constant collision energy of 2 eV in both Trap cell and Transfer cell. In high-energy HDMSE mode, the collision energy was ramped from 25 to 55 eV in the Transfer cell only. The RF applied to the quadrupole mass analyzer was adjusted such that ions from m/z 300 to 2000 were efficiently transmitted, ensuring that any ions observed in the LC-HDMSE data <m/z 300 arose from dissociations in the Transfer collision cell. The acquired LC-HDMSE data were processed and searched against protein knowledge database (Uniprot) through ProteinLynx Global Server (PLGS, Waters Company) with 4% FDR.

For cell line subcellular and SDS/PAGE fractionated samples, trypsin digested peptides were separated by reversed-phase chromatography using an EASYnano HPLC system (Thermo Scientific) coupled online with a LTQ-Orbitrap ELITE mass spectrometer (Thermo Scientific). The system was equipped with a WATERS Symmetry C18 nanoAcquity trap-column (180 μm × 20 mm, 5 μm) and a C18 analytical column 75 μm × 250 mm, 3 μm, (Column Technology Inc.). The separation column temperature was set ambient, and the temperature of the tray compartment in the auto-sampler was set at 6 °C. Mass spectrometer parameters were spray voltage 2.5 kV, capillary temperature 280 °C, FT resolution 60,000, FT target value $1 \times 10^6$, LTQ target value $3 \times 10^4$, 1 FT microscan with 500 ms injection time, and 1 LTQ microscan with 10 ms injection time. Mass spectra were acquired in a data-dependent mode with the m/z range of 400–2000. The full mass spectrum (MS scan) was acquired by the FT and tandem mass spectrum (MS/MS scan) was acquired by the LTQ with a 35% normalized collision energy. Acquisition of each full mass spectrum was followed by the acquisition of MS/MS spectra for the 20 most intense +2 or +3 ions within a one second duty cycle. The minimum signal threshold (counts) for a precursor occurring during a MS scan was set at 5000 for triggering a MS/MS scan. The acquired LC-MS/MS data was processed by the Proteome Discoverer 1.4 (Thermo Scientific). The Sequest HT was used as a search engine with the parameters including cysteine (Cys) alkylated with acrylamide (71.03714@C) as a fixed modification and methionine (Met) oxidation (15.99491@M) as a variable modification. Mass precursor of ±20 ppm and MS2 fragment of ±0.7 Da were used for the mass tolerance. Data was searched against the Uniprot Human database November 2015 and further filtered the data with FDR ≤ 5%. Total MS counts for each protein were used as a measure of protein abundance.

The HPLC fractionated HLA-bounded peptides were analyzed by LTQ-Orbitrap ELITE mass spectrometer (Thermo Scientific). The LC/MS system and parameters were same besides the acquisition of each full mass spectrum was followed by the acquisition of MS/MS spectra for the 20 most intense including +1, +2 and +3 ions within a one second duty cycle. For peptide identification, Proteome Discoverer 1.4 was used and amino acid length of 12–34 was searched with no enzyme digestion against the Uniprot database separately. The mass tolerance, modifications and FDR were performed as described above. Peptides of 12- to 34-aa residues were subsequently filtered based on the fitness score for the consensus binding motifs for each of the HLA-II alleles, according to the Immune Epitope Database (www.iedb.org). Predicted HLA class II-bound peptides were further inspected for mass accuracy and MS/MS spectra were validated manually. All predicted HLA class II peptides were searched using the NCBI BLASTp algorithm against the human non-redundant protein database to identify the peptide corresponding source.

**Luminex analysis**. An equal number of exosomes isolated from six PDAC cell lines (CFPAC-1, HPAF-II, SU.86.86, Panc 03.27, MIA PaCa-2, and PANC-1), as quantified by nanoparticle-tracking analysis, were pooled and were coupled to xMAP carboxylated microspheres following the manufacturer's protocol (Luminex Corporation, Austin TX). The optimal concentration of exosomes (32 μg per 10$^6$ beads) was determined by titration in the 0.5–64 μg per 10$^6$ beads range using anti-CD63 (sc-5275, Santa Cruz) and anti-TSG101 (ab83, Abcam) antibodies. Similarly, a distinct set of xMAP carboxylated microspheres were directly coupled with 5 μg

per 10$^6$ beads anti-human Ig (The Jackson Laboratory) and applied to normalize the autoantibody signal. A set of uncoated beads were used for assessment of non-specific binding. Equivalent counts of each set of protein-coupled microspheres were mixed to a concentration of 5000 per set per 50 μL per well in PBS containing 10% normal goat serum. The beads were incubated with 50 μL patient serum diluted 1:100 in a 96-well filter-bottomed microtiter plate on a shaker for 1 h in the dark at RT. The beads were washed thrice and then incubated with 2 μg per mL of biotin-conjugated goat anti-human IgG (The Jackson Laboratory) for 1 h in the dark. After three washes, the beads were incubated with 4 μg per mL of Strepavidin-Phycoerythrin (SAPE) for 30 min at RT in the dark. After the final washing step, beads were resuspended in 100 μL of MAGPIX® drive fluid (Luminex Corporation) and their median fluorescence intensity (MFI) was read on the MAGPIX® instrument (Luminex Corporation). Exosome-coupled bead MFI was normalized on the anti-human Ig bead MFI and expressed as: log$_2$(MFI$_{exosome\ beads}$/MFI$_{uncoated\ beads}$).

**Complement-dependent cytotoxicity assay**. Exosomes applied in CDC assays were isolated using the density gradient floatation approach described above and underwent buffer exchange (in serum free RPMI) using Zeba™ Desalt Spin Columns according to manufacturer's instructions (Thermo Scientific). A pool of pre-diagnostic PDAC patient and matched healthy control (n = 13 per each group; set #2 in Supplementary Table 6) serum samples from the CARET study were used to mediate the CDC.

To test the dose-dependent decoy function of PDAC exosome in inhibiting CDC mediated by PDAC serum, PANC-1 cells were seeded in a 96-well plate (5 × 10$^3$ per well) in RPMI + 1% FBS overnight for adhesion. Cells were washed with 200 μL pre-warmed serum free RPMI, incubated in presence of increasing amount of exosomes (0, 1, and 2.5 μg; 25 μL per well) and with equal volume of sera diluted 1:10 in PBS for 15 min at RT, followed by incubation with 50 μL of fresh reconstituted rabbit complement (Low-Tox rabbit complement; Cedarlane) for 2 h at 37 °C. Total lysis of the cells was achieved by solubilizing a non-treated control sample with 4% CHAPS. Antibody-independent (spontaneous) lysis was controlled for each experimental condition (0, 1, and 2.5 μg of exosomes) by incubating cells with complement and without serum in presence of equal amount of exosomes. Viability was evaluated with the CellTiter 96 Aqueous One Solution (Promega). The percentage of cell viability was calculated as: (experimental maximum/spontaneous maximum) × 100.

Live cell imaging cytotoxicity assay was performed to test in real time the decoy function of PDAC or non-neoplastic exosomes in inhibiting CDC mediated by PDAC or matched healthy control sera. PANC-1 or Pa03C cells were seeded in a 96-well plate (10 × 10$^3$ per well) in RPMI + 1% FBS overnight for adhesion. Cells were washed with 200 μL pre-warmed serum free RPMI, incubated in presence or absence of exosomes (5 μg per well; 50 μL per well) with equal volume of sera diluted 1:2 in PBS for 30 min at RT. Cells were washed with 200 μL serum free RPMI, incubated with 50 μL per well of IncuCyte™ Cytotox Green Reagent (Essen Bioscience) diluted 1:1000 in serum free RPMI, followed by incubation at 37 °C with equal volume of fresh reconstituted rabbit complement (Cedarlane). Green fluorescence (dead cells) and cell confluency was measured on an Incucyte™ ZOOM instrument (Essen Bioscience), and the number of green dead cells was calculated using ZOOM software (Essen Bioscience). To control for antibody-independent (spontaneous) lysis, the number of green dead cells obtained by incubating the cells with complement and without serum was subtracted to each experimental condition. All measurements were performed in triplicates.

**Recombinant protein array production, staining and imaging**. Recombinant Protein antigens (Life Technologies) were resuspended in printing buffer (250 mM Tris-HCl, pH 6.8; 0.5% sodium dodecyl sulfate; 25% glycerol; 0.05% TritonX-100; 75 mM dithiothreitol) and spotted at a concentration of 1 mg mL$^{-1}$ onto 16-pad nitrocellulose-coated slides (Maine Manufacturing) using a contact printer (Genetix QArray2). Printing buffer was spotted as negative control, purified human IgG and EBNA as positive controls. Plasma samples were individually hybridized on each slide at a dilution of 1:150. Secondary antibody reaction with a fluorescent anti-human IgG (1 μg mL$^{-1}$, DyLight 649-conjugated rabbit anti-human IgG, Jackson ImmunoResearch) was performed. The protein arrays were scanned with a GenePix scanner (Molecular Devices GenePix Pro 4000B) using a red laser (635 nm) at constant detector gain and power setting for all slides. Local background-subtracted median spot intensities were generated using GenePix Pro 7 Software and used for downstream statistical analysis.

**Analysis of proteomic data**. Mass spectrometry data of Ig-bound, cell line and exosome proteins underwent quantile normalization using the normalize.quantiles function from the R/Bioconductor package preprocessCore (version 1.40.0) to make the distributions the same across samples. Quantile normalization does not force means of samples to converge upon a predetermined value, as has been observed in other normalization techniques, i.e., central tendency, linear regression and local regression), where the mean of peptide abundances can be forced to converge to zero[42]. Cell line and exosome surface proteins were selected as follows: >5 normalized MS2 counts in at least two of six cell lines, >1.25-fold change in average number of normalized MS2 counts in surface versus both cargo and TEE compartments for

exosomes and versus TCE compartment for cells. A total of 308 proteins identified in the Ig-bound fractions with at least 5 normalized MS2 spectral counts were selected for downstream analysis. Ninety-two proteins were selected from this list based on the following criteria: (i) a case-to-matched control average MS2 count ratio of 1.5 or greater; and (ii) confirmed expression of the corresponding genes in a panel of 11 PDAC cell lines, as well as in The Cancer Genome Atlas (TCGA) PDAC dataset ($n = 112$ patients). We excluded from the list of 92 proteins immunoglobulin chains, acute phase response proteins annotated in Ingenuity Pathway Analysis (IPA) Software (Ingenuity Systems, Mountain View, CA), and the most abundant plasma proteins[43]. For downstream analysis of plasma exosome proteomics data, only proteins identified in at least two samples were selected. Genesis software (genome.tugraz.at/genesisclient/genesisclient_description.shtml)[44] was used to perform complete linkage hierarchical clustering of proteomics data; Pearson correlation was applied as the basis for distance measure. Enrichment pathway analysis was performed using IPA Software and KEGG pathway WEB-based GEne SeT AnaLysis Toolkit (www.webgestalt.org). GO localization analysis was performed using the MetaCore software (clarivate.com/products/metacore/).

**Gene expression analysis.** mRNA extraction was performed by using RNeasy Maxi Kit (Qiagen). RNA-Seq libraries were prepared by using standard Illumina reagents and protocols. Paired-end sequencing with the read length of 50 bases were performed on the Illumina HiSeq 2000 platform following the manufacturer's instructions. CASAVA 1.8.2 was used to demultiplex and to generate FASTQ files. The sequencing data were analyzed by using the software packages of Tophat mapping against hg19 and Cufflinks[45]. The number of reads was normalized and expressed as fragments per kilobase of exon per million fragments mapped (FPKM).

**The Cancer Genome Atlas RNA-seq data analysis.** Level 3 RNA-seq data generated using the Illumina HiSeq 2000 RNA Sequencing Version 2 platform were downloaded for pancreatic (112 PDAC) cancer patients from the The Cancer Genome Atlas Data Portal (portal.gdc.cancer.gov/projects/TCGA-PAAD). The individual RNA-seq sample expression files were compiled into a dataset using R v3.1.1 and R-studio v0.98.1062. Gene expression level was determined using log2 transformed RSEM normalized read counts.

**Statistical analysis.** Categorical data were compared by Fisher's exact test. Student's t-test and Mann–Whitney t-test was used to assess the differences in continuous variables, as appropriate. Hypergeometric test was applied to calculate the statistical significance of the overlap between two gene lists and the corresponding representation factor (using nemates.org/MA/progs/overlap_stats.html, where the total number of genes in the genome is set at 17,611). Receiver operating characteristic curve analysis was performed to assess the performance of autoantibody in distinguishing PDAC cases from healthy controls and chronic pancreatitis cases. In the protein array analysis, owing to the small sample size of each set, sets #1, #2, and #3 were merged by standardizing the data such that the mean was 0 and standard deviation was 1 for healthy controls. Statistical analysis was done using R (version 3.2.3) and GraphPad Prism software (Version 6.0). We used a two-sided significance level of 0.05 for all statistical analyses, unless a direction of the difference was applied expected as specified. p-values of <0.05 were considered statistically significant for all tests. Where appropriate, fig. legends define n, which indicates the number of individual biological replicates.

## Data availability

The MS proteomics data generated and analyzed during this study have been deposited to the ProteomeXchange Consortium via the PRIDE[46] partner repository with the dataset identifier PXD011957. RNA-seq datasets used for gene expression analyses in this study have been deposited to the NCBI Gene Expression Omnibus (GEO) repository and are available through accession code GSE122818. The TCGA PDAC data referenced during the study are available from the TCGA public Data Portal (portal.gdc.cancer.gov/projects/TCGA-PAADcancergenome.nih.gov). Other relevant data supporting the findings of this study are available within the Article and Supplementary Files, or are available from the authors upon reasonable request.

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

## Acknowledgements

This work was supported by the MD Anderson Moonshot Program, NIH National Cancer Institute Early Detection Research Network grant U01 CA200468 and MCL Consortium grant U01 CA196403, the Pancreatic Cancer Action Network, and a faculty fellowship from The University of Texas MD Anderson Cancer Center Duncan Family Institute for Cancer Prevention and Risk Assessment. The High Resolution Electron Microscopy Facility at UTMDACC is supported by CCSG grant NIH P30 CA016672. We acknowledge Kenneth Dunner, Jr. (UTDMACC) for providing valuable technical assistance with electron microscopy sample preparation and image acquisition.

## Author contributions

M.C., J.V.V., A.M., A.T., and S.M.H. designed experiments, performed data analyses, and wrote the manuscript. M.C., J.V.V., D.L.K., M.A., D.S.D., S.C.T., S.F.-B, H.A., V.B., and H.P. designed and performed experiments. C.A.-B. and A.A.M. performed bioinformatics analyses. L.E.B. and Z.F. performed biostatistical analysis. H.K. and H.W. performed mass spectrometry analysis and wrote the manuscript. M.H.K, R.B., D.G.A., M.A.F., S.J.M., and J.J.M. provided biospecimens, performed data analyses, and wrote the manuscript.

## Additional information

**Competing interests:** The authors declare no competing interests.

