## [Peer Review File · Nature Communications]

Reviewers' comments:

Reviewer #1: PDAC immunology expert
(Remarks to the Author):

This is a nice study investigating a novel finding of antibody responses to cancer derived exosomes in PDAC patients. The use of a sophisticated mass spectrometric analyses on number of clinical samples in various cancer contexts is quite exciting. The primary question that the study can answer more definitively is whether these antibody responses are exosome-specific or cancer exosome-specific. The prospective demonstration of these responses (antibody responses to cancer-exosomes, presence of cancer exosomes, tumor antigen targets on cancer exosome with bound antibody) all in individual patients will greatly enhance the rigor and impact of the study.

Major points:

1. The purity assessments of Ig-bound proteins are unclear. Can the authors demonstrate that the extracted proteins from the protein agarose assays are truly Ig bound, and are not contaminated with exosomes? This is a critical point as if the extraction is impure, there can be contamination of the "Ig-bound proteins" with exosomal proteins, which would then confound all the downstream analysis.
2. What is the rationale for selecting only a fraction of the Ig-bound proteins for analysis? The authors should show both the unbiased (without selection) and the selected analyses.
3. Both the rationale and methodology of selection process by which the authors pick the 92 proteins from the greater pool of 308 can be clarified a bit. Specifically
 - a. What is the case-control matching? Stage? Treatment?
 - b. What is the rationale for a count ratio of > 1.5 ?
 - c. The confirmed expression of the TCGA data should be shown.
 - d. Suppl. Tables 2 and 3 are missing
4. Figure 1C – this does not allow an interpretation of the relative amounts of these proteins in PDAC-pts vs. controls. The relative amounts in the controls should be shown so that quantitative comparisons can be made.
5. How do the authors conclude that Ig-bound proteins are enriched in exosome-associated proteins (line 86)? In Fig 1C, only 6/22 of the gene sets are "exosome-associated" by the authors definitions.
6. To definitely conclude that Ig-bound proteins are enriched in exosomal proteins, can the authors examine the relative abundance of exosome-specific markers in Ig-bound fractions in PDAC vs. control – eg., glypican 1 and/or other exosomal markers?
7. Figure 2D – The n's are small and it is a bit difficult to make definitive conclusions. Can the authors increase the n's here for a more robust assessment?
8. Figure 3 should be performed on patient-derived cancer exosomes, not just on cell lines. Can the authors show that proteomic profiling can identify presence of tumor antigens on cancer exosomes, as enriched by markers such as Glypican 1 and/or others?
9. Suppl. Figure 3C should include a biological control – i.e., are these exosome surface enriched proteins that the authors find (n=77, Suppl. Fig 3C) also enriched in exosomes in healthy controls? This would be important to help distinguish whether these findings reflect a cancer-specific enrichment or on the contrary, an exosome specific enrichment in both non-cancer and cancer contexts.

10. What are the biological controls for Figure 6C? The authors should show that sera from normal patients does not exert similar effects as PDAC sera, to demonstrate specificity of the findings to cancer exosomes.

Minor points:

1. Figure 1: The n's should be clearly indicated on the figures – eg: in Fig. 1B.

In general, a flow chart indicating what patients were used for what analyses and how many of the analyses utilized the same patients will be very helpful in enhancing the clarity.

2. The tables in Suppl Table 1 are appearing jumbled and are a bit difficult to interpret. This can be made a bit clearer.

3. Figure 2: the relative differences between particle size in healthy and PDAC pts should be quantified with statistical analyses.

Reviewer #2 Expert in Exosomes and cancer immunology

(Remarks to the Author):

This study describes a very interesting new concept, that cancer-cell specific antibodies are neutralized by binding to cancer-cell derived exosomes in the circulation. This can potentially have a high impact on both biomarker analysis and the development of new cancer therapies. In fact, I think that the authors are too modest when discussing the implications for novel treatment strategies.

The authors start with clinical plasma samples from patients with pancreatic ductal adenocarcinoma (PDAC). In-depth proteomic profiling by mass spectrometry of immunoglobulin-bound proteins in plasmas showed the identification of tumor antigens that induce antibody response as well as exosomal proteins. This demonstrates that exosome-bound Igs are present in serum of PDAC patients. They further show that exosomes from plasma of patients and cell lines have overlapping antigenic proteins. They conclude that exosomes function as decoy receptors for cancer-specific antibodies. This have major implications for how tumor derived exosomes contribute to tumor immune escape.

The study is very thorough and well preformed, and I only have a few comments.

1. In figure 6, PDAC-derived exosomes induced a dose-dependent inhibition of PDAC serum mediated complement-dependent cytotoxicity directed at cancer cells. This experiment needs to be repeated with an unrelated exosome. This would show if this is antigen-specific or not. There are other molecules in exosomes that also could increase the survival, so this needs to be clarified. I don't expect other exosomes to have no effect, but it should be lower.

2. In the discussion, the results are mostly discussed around the role of exosomes as potential biomarkers. The decoy function, and how this could be explored in future treatment strategies, should be discussed more.

Reviewer #3: Expert in humoral responses

(Remarks to the Author):

The authors identified auto-antigens (TAAs) bound to serum antibodies from pancreatic ductal adenocarcinoma (PDAC) patients. Additionally they isolated exosomes from PDAC patients and cell lines and showed that they bind to au-antibodies from patients. Using additional LC-MS/MS

analyses they identified a number of proteins that were enriched on exosome surfaces several of which were also detected as soluble TAAs. Finally, the authors present evidence that patient derived PDAC exosomes inhibit CDC of pancreatic cell lines and conclude that "they exert a decoy function against complement-mediated cytotoxicity". Overall there is no doubt that the authors performed a significant amount of work and some of their findings are quite interesting and thus could represent an advance in the field of sufficient interest to warrant publication in NC. However, in part because of the complexity of the experiments there are many technical issues that need to be addressed. Also some of the data appear to be have been over interpreted and critical controls have not been provided. Thus it is recommended that the authors perform extensive revisions before the manuscript is considered further.

Major: 1) A key conclusion of the manuscript as mentioned above is that exosomes may serve as a decoy function in CDC with auto-antibodies. This is based on the data in Fig 6C. First off critical controls are missing: The exosomes may be inhibiting CDC not because of a decoy effect but rather because they contain complement inhibitory proteins. This in fact is a much more likely explanation. If that is the case then exosomes should inhibit non-specifically . Second, the authors used the PANC-1 cell line which in many ways differs substantially from patient derived PDAC cells. To support their key conclusion the authors need to at the very least repeat this experiment with other cell lines. Third a control showing CDC of PANC-1 (and other cell lines) with unrelated sera should be shown -it is not even clear from this experiment that CDC was due to the activation of the classical pathway! I can go on, but the point here is that this is a major conclusion highlighted in the abstract and it has to be supported by rigorous data.

2) TAA identification from patient sera: The authors used Protein G/A to isolate antibodies with their bound antigens. Critical details for how this experiment was performed need to be provided: Obviously in such an experiment there should be a huge excess of pulled down IgG relative to bound antigens. The complexity of the starting mixture i.e. sera can introduce a number of complications: Did the authors wash the Protein A/G column to reduce the amount of non-specific proteins that might be bound onto the column. A low pH wash (ph 5-6) is usually required. In such experiments one is typically likely to find primarily "sticky proteins" that non-specifically associate with antibodies and also very abundant cellular proteins. A list of the 308 TAA proteins detected should be provided. When the authors say they report on proteins detected with at least 5 "normalized MS2 Spectral counts they need to explain how the normalization was performed.

3) Importantly, there is no mention of how many proteins were identified in the sera of multiple donors.

4) The authors patients with and without T2D (Fig 1B) but then do not report on any differences between the two groups nor do they say why analyzing the two groups is relevant in the context of PDAC.

5) Fig 3C: line 114 in the text indicates that the data in that figures were derived from MS2 counts whereas the Fig legend refers to intensity. There are several reasons why quantitative conclusions should not be drawn from MS2 counts. Have the authors looked at the respective MS1 intensities and do the data match? In any event the quantitative analysis needs to be supported by additional independent evidence.

6) Same concerns apply to Fig 4B and C. What do the different data in the heat map in Fig 4C correspond to?

7) The high level of intracellular proteins found on exosomes is a concern. It could be that simply apoptotic debris aggregates and accumulates non-specifically on endosome surfaces. How confident were these IDs? Were they based on multiple peptides a fact that would enhance the likelihood that the intact protein and not proteolytic fragments somehow ended up on the surface of exosomes.

8) Line 279: These data provide evidence of a direct binding of autoantibodies to intact exosomes in contrast to free circulating proteins," I do not see how this conclusion is supported by the data!

Minor:

1) Line 290; "As B lymphocytes are able to recognize conformational epitopes, our findings

suggest that, exosomes must expose B cell epitopes as native antigens." This does not follow in any way from the data. BCRs can recognize either peptides or conformational epitopes!

Reviewer 1: *This is a nice study investigating a novel finding of antibody responses to cancer derived exosomes in PDAC patients. The use of a sophisticated mass spectrometric analyses on number of clinical samples in various cancer contexts is quite exciting. We thank the Reviewer for commending us on our study.*

1. *The purity assessments of Ig-bound proteins are unclear. Can the authors demonstrate that the extracted proteins from the protein agarose assays are truly Ig bound, and are not contaminated with exosomes? This is a critical point as if the extraction is impure, there can be contamination of the “Ig-bound proteins” with exosomal proteins, which would then confound all the downstream analysis.*

The Reviewer raises a valid concern. We now provide additional details regarding the protocol used for Ig bound protein isolation in the *Methods* section. In brief, we applied Nab Protein A/G spin columns (Thermo Scientific) and followed the manufacturer’s recommended procedure; to mitigate nonspecific binding to the column, samples were diluted in binding buffer and the column was washed extensively before and after sample loading. Given the presence of intact exosomes in the Ig bound fraction (Figure 2 and 3) we tested, as suggested by the Reviewer, the extent of nonspecific binding of exosomes to the Protein A/G column using PDAC and healthy control plasma samples. As shown in Supplementary Figure 1, we compared the exosome yield, as quantified by particle counting, using the standard protocol we previously applied as well as after performing an additional low pH wash (pH 5) after plasma sample loading. We observed that particle recovery was not significantly changed after low pH wash and that minimal exosome particle counts were detected in flow-through from the low pH wash. These data suggest negligible passive binding of exosome particles to the protein A/G column. Moreover, binding of exosomes to immunoglobulins in circulation has been further verified as reported in Figures 2 and 3.

2. *What is the rationale for selecting only a fraction of the Ig-bound proteins for analysis? The authors should show both the unbiased (without selection) and the selected analyses.*

In order to enrich for tumor associated antigen candidates in the Ig-bound fraction, we selected for downstream analysis Ig-bound proteins that exhibited comparatively higher levels in PDAC patients compared to matched controls. We now provide in Supplementary Table 2 the complete list of 308 identified proteins and also report the percentage of PDAC and control samples in which each protein was identified, the ratio of MS2 counts in PDAC cases relative to matched controls and the mRNA expression levels of individual proteins in PDAC tissues (TCGA dataset) and pancreatic cancer cell lines.

3. *Both the rationale and methodology of selection process by which the authors pick the 92 proteins from the greater pool of 308 can be clarified a bit.*

Please see above and also response to Reviewer 3, question 2, for additional details regarding filtering of the Ig-bound protein data.

a. *What is the case-control matching? Stage? Treatment?*

In order to control for possible confounding factors in the two analyzed cohorts, patients and controls were matched based on gender, age and T2D status (please also see answer to Reviewer 3, question 4) . This information, together with stage, is provided in Supplementary Table 1. The analyzed patients were treatment naïve, as now specified in the *Methods* section.

b. *What is the rationale for a count ratio of > 1.5?*

Ratio distribution shows a median ratio of 0.947 (95%CI: 0.891-1.0); the applied count ratio threshold of 1.5 corresponds to the 65th percentile of the dataset. This filtering criterion

yielded 92 candidates and was chosen in order to maximize the initial pool of tumor associated antigen candidates which then underwent subsequent downstream validation. Given the small sample size, application of statistical filtering is overly restrictive; secondary selection criteria were based on gene expression in PDAC cell lines and the PDAC TCGA datasets. Details regarding these filtering criteria are now described in the *Methods* section. A similar approach has been applied in previous discovery studies published by our group (Taguchi A. *et al*, Cancer Cell 2011 and Capello M. *et al*, Journal of the National Cancer Institute 2017)

c. *The confirmed expression of the TCGA data should be shown.*

As suggested, confirmed expression of the corresponding genes in a panel of 11 PDAC cell lines, as well as in The Cancer Genome Atlas PDAC dataset is reported in Supplementary Tables 3 and 4.

d. *Suppl. Tables 2 and 3 are missing*

We apologize for the confusion regarding these Supplementary Tables. We have confirmed that they are included in the current Supplementary Excel file.

4. *Figure 1C – this does not allow an interpretation of the relative amounts of these proteins in PDAC-pts vs. controls. The relative amounts in the controls should be shown so that quantitative comparisons can be made.*

We agree with the Reviewer that this additional data would be informative. The relative amounts of the selected 308 and 92 Ig-bound proteins are now reported in Supplementary Tables 2, 3 and 4.

5. *How do the authors conclude that Ig-bound proteins are enriched in exosome-associated proteins (line 86)? In Fig 1C, only 6/22 of the gene sets are “exosome-associated” by the authors definitions.*

MetaCore pathway analysis of the Ig-bound protein data set revealed the top four cellular localizations represented were *extracellular exosome*, *extracellular vesicle*, *extracellular organelle*, and *vesicle*, exhibiting p-values $< 8.44 \times 10^{-11}$ and FDRs $< 8.29 \times 10^{-9}$ and indicating distinct and significant representation of exosome and extracellular vesicle associated proteins within the Ig-bound plasma fraction.

6. *To definitely conclude that Ig-bound proteins are enriched in exosomal proteins, can the authors examine the relative abundance of exosome-specific markers in Ig-bound fractions in PDAC vs. control – eg., glypican 1 and/or other exosomal markers?*

Mass-spectrometry-based quantitation of integral membrane proteins such as tetraspanins represents a field-wide challenge. Indeed, in our analysis we did not detect classical exosomal markers; this could be related to the low abundance and high lipophilicity of these molecules. However, among the 92 proteins exhibiting higher levels in the Ig-bound fraction of PDAC patients compared to matched controls, we have identified annotated vesicular trafficking and biogenesis molecules such as RAN, ARF6, Endofin (ZFYVE16), TMEM175, ATP9B, and ITPR2. These data have been added to the *Results* section.

7. *Figure 2D – The n’s are small and it is a bit difficult to make definitive conclusions. Can the authors increase the n’s here for a more robust assessment?*

We agree with the Reviewer that adding more samples would improve this assessment. The input volume requirement limits availability of samples for these experiments; nevertheless,

we have now increased our cohort to 7 samples per group and provide the respective statistical analysis in Figure 2D.

8. *Figure 3 should be performed on patient-derived cancer exosomes, not just on cell lines. Can the authors show that proteomic profiling can identify presence of tumor antigens on cancer exosomes, as enriched by markers such as Glypican 1 and/or others?*

The data presented in Figure 3 is restricted to proteins which were identified by mass-spectrometry profiling of plasma-derived exosomes from PDAC patients and matched controls. Supplementary Table 5 shows robust representation of proteins identified in the plasma-derived exosome samples that are involved in exosome biogenesis, vesicular trafficking, and cytoskeletal regulation, as well as exosome cargo enriched proteins annotated in the Exocarta database. Together, these confirm high purity and yield of our exosome preparations. Figure 5D shows the enrichment of tumor associated antigens able to induce autoantibodies in PDAC in exosomes isolated from patient plasma compared to matched controls.

9. *Suppl. Figure 3C should include a biological control – i.e., are these exosome surface enriched proteins that the authors find (n=77, Suppl. Fig 3C) also enriched in exosomes in healthy controls? This would be important to help distinguish whether these findings reflect a cancer-specific enrichment or on the contrary, an exosome specific enrichment in both non-cancer and cancer contexts.*

We agree with the reviewer that some of the observed proteins (e.g. in Supplementary Figure 5B) are not strictly cancer-related. However, they are known to be involved in triggering an autoantibody response in the context of autoimmune diseases, such as systemic lupus erythematosus. It is logical to consider that exosomes might be involved in triggering an autoantibody response also in autoimmunity, as we comment on in the *Discussion*.

Nevertheless, the 77 tumor-associated antigens highlighted in Supplementary Figure 5C have been previously reported to induce autoantibodies across different tumor types, as annotated in the Cancer Immunome Database. Moreover, as described above, we identified higher levels of tumor-associated antigens in circulating plasma exosomes isolated from PDAC patients relative to matched controls (Figure 5D).

10. *What are the biological controls for Figure 6C? The authors should show that sera from normal patients does not exert similar effects as PDAC sera, to demonstrate specificity of the findings to cancer exosomes.*

We agree with the Reviewer. We now compare, using two different cell lines, the inhibitory effect of PDAC exosomes against complement-dependent cytotoxicity mediated by either PDAC patient or matched healthy control sera. We confirmed a significant effect of PDAC exosomes in exerting a decoy function against CDC mediated by PDAC but not healthy control samples (Figure 6D and Supplementary Figure 11). Please also see the response to Reviewer 2, question 1 and to Reviewer 3, question 1.

Minor points:

1. *Figure 1: The n's should be clearly indicated on the figures – eg: in Fig. 1B.*

In general, a flow chart indicating what patients were used for what analyses and how many of the analyses utilized the same patients will be very helpful in enhancing the clarity.

Figure 1B has been modified as suggested. We have also included in Supplementary Table 1 and 8 a summary of all the applied cohorts, patient characteristic information and the types of analyses performed using each sample set.

2. *The tables in Suppl Table 1 are appearing jumbled and are a bit difficult to interpret. This can be made a bit clearer.*

Supplementary Table 1 has been modified as suggested.

3. *Figure 2: the relative differences between particle size in healthy and PDAC pts should be quantified with statistical analyses.*

This analysis is now provided in Figure 2B and Supplementary Figure 2.

Reviewer 2: *This study describes a very interesting new concept, that cancer-cell specific antibodies are neutralized by binding to cancer-cell derived exosomes in the circulation. This can potentially have a high impact on both biomarker analysis and the development of new cancer therapies. In fact, I think that the authors are too modest when discussing the implications for novel treatment strategies.*

We thank the reviewer for the supportive assessment regarding the relevance of our work.

The study is very thorough and well preformed, and I only have a few comments.

1. *In figure 6, PDAC-derived exosomes induced a dose-dependent inhibition of PDAC serum mediated complement-dependent cytotoxicity directed at cancer cells. This experiment needs to be repeated with an unrelated exosome. This would show if this is antigen-specific or not. There are other molecules in exosomes that also could increase the survival, so this needs to be clarified. I don't expect other exosomes to have no effect, but it should be lower.*

As suggested by the Reviewer, we have tested the effect of exosomes isolated from an immortalized PDAC-associated fibroblast line (CAF19) in attenuating complement-dependent cytotoxicity. We demonstrated no inhibitory effect of non-neoplastic CAF19 exosomes against CDC mediated by PDAC sera (Figure 6D). Please also see response to Reviewer 1, question 10 and to Reviewer 3, question 1.

2. *In the discussion, the results are mostly discussed around the role of exosomes as potential biomarkers. The decoy function, and how this could be explored in future treatment strategies, should be discussed more.*

We have expanded comment on the therapeutic implications of exosome decoy function in the *Discussion* section.

Reviewer 3: *[...] Overall there is no doubt that the authors performed a significant amount of work and some of their findings are quite interesting and thus could represent an advance in the field of sufficient interest to warrant publication in NC.*

We thank the Reviewer for the interest in our study.

1) *A key conclusion of the manuscript as mentioned above is that exosomes may serve as a decoy function in CDC with auto-antibodies. This is based on the data in Fig 6C. First off critical controls are missing: The exosomes may be inhibiting CDC not because of a decoy effect but rather because they contain complement inhibitory proteins. This in fact is a much more likely explanation. If that is the case then exosomes should inhibit non-specifically. Second, the authors used the PANC-1 cell line which in many ways differs substantially from patient derived PDAC cells. To support their key conclusion the authors need to at the very least repeat this experiment with other cell lines. Third a control showing CDC of PANC-1 (and other cell lines) with unrelated sera should be shown -it is not even clear from this experiment that CDC was due to the activation of the classical pathway! I can go on, but the point here is that this is a major conclusion highlighted in the abstract and it has to be supported by rigorous data.*

Additional detail regarding the protocol applied for the CDC assay reported in Figure 6C has been provided in the *Methods* section. Moreover, to address the Reviewer's concerns we have performed additional CDC assays by live cell imaging cytotoxicity analysis (IncuCyte, Essen BioScience; see *Methods* for details) to allow for sensitive, dynamic assessment of autoantibody mediated cytotoxicity. In order to control for possible confounding factors and exclude any non-specific inhibition of CDC mediated by exosomes we have also improved our CDC assay protocol. After a 30 minutes incubation of cells with patient or control sera in presence or absence of exosomes, cells were washed and then the complement was added. We controlled for cytotoxic effects induced by the alternative complement pathway by subtracting the spontaneous (antibody-independent) cytotoxicity mediated against cancer cells by complement alone in absence of serum. The decoy function of PDAC exosomes against complement-dependent cytotoxicity mediated by healthy control sera has been tested and no significant effect has been observed (Figure 6D and Supplementary Figure 11). Finally, to confirm our hypothesis, we performed CDC assays using a PDAC cell line established from metastatic tissue (Pa03C) and exosomes isolated from its conditioned media (Figure 6D). Please also see response to Reviewer 1, question 10 and to Reviewer 2, question 1.

2) *TAA identification from patient sera: The authors used Protein G/A to isolate antibodies with their bound antigens. Critical details for how this experiment was performed need to be provided: Obviously in such an experiment there should be a huge excess of pulled down IgG relative to bound antigens. The complexity of the starting mixture i.e. sera can introduce a number of complications: Did the authors wash the Protein A/G column to reduce the amount of non-specific proteins that might be bound onto the column. A low pH wash (ph 5-6) is usually required. In such experiments one is typically likely to find primarily "sticky proteins" that non-specifically associate with antibodies and also very abundant cellular proteins. A list of the 308 TAA proteins detected should be provided. When the authors say they report on proteins detected with at least 5 "normalized MS2 Spectral counts they need to explain how the normalization was performed.*

We appreciate the Reviewer's careful consideration of our experimental approach. Please see response to Reviewer 1, question 1 for details concerning the Ig-bound protein isolation protocol and assessment of non-specific binding of exosomes to the protein A/G column. Additionally, the MS profiling data sets were filtered to exclude immunoglobulin chains, acute phase response proteins annotated in Ingenuity Pathway Analysis (IPA) Software, and the most abundant plasma proteins (Anderson NL and Anderson NG, *Molecular & cellular proteomics* 2002) from our list of 92 proteins enriched in PDAC patient Ig bound fractions compared to matched controls. The description of the normalization procedure and Ig-bound protein selection is now detailed in the *Methods* section. As described above, we now also provide in Supplementary Table 2 the complete list of the 308 Ig bound proteins. We would like to emphasize that multiple downstream analyses (i.e. see Figure 5 and 6) have been performed to confirm the antigenicity of exosome surfaceome protein subsets. Moreover, circulating plasma exosomes applied during downstream validation experiments (i.e. Figure 2, Figure 3 and Figure 5D) have been isolated using a single gradient centrifugation approach (see *Methods*) which has been optimized for depletion of abundant free proteins.

3) *Importantly, there is no mention of how many proteins were identified in the sera of multiple donors.*

We now report in Supplementary Table 2 the percentage of PDAC and control samples in which each protein was identified, the ratio of MS2 counts in PDAC cases over matched controls and the mRNA expression levels of individual proteins in PDAC tissues (TCGA dataset) and cell lines. Please also see response to Reviewer 1, question 2.

4) *The authors patients with and without T2D (Fig 1B) but then do not report on any differences between the two groups nor do they say why analyzing the two groups is relevant in the context of PDAC.*

T2D has been considered in this analysis only as a matching factor between cases and controls. T2D is frequently associated with PDAC and as a pro-inflammatory condition it might represent a possible confounding factor in the analysis. This point is now clarified in the *Methods* section. Ig-bound protein MS2 count ratio in PDAC cases relative to matched controls for individual groups is listed in Supplementary Table 2 and 3.

5) *Fig 3C: line 114 in the text indicates that the data in that figures were derived from MS2 counts whereas the Fig legend refers to intensity. There are several reasons why quantitative conclusions should not be drawn from MS2 counts. Have the authors looked at the respective MS1 intensities and do the data match? In any event the quantitative analysis needs to be supported by additional independent evidence.*

This is a mistake; the data reported in Figure 3C indeed indicate intensity. We have now corrected the text in the *Results* section. For profiling of cell line derived exosomes where the sample input is relatively unlimited, MS2 counts are a good representation of peptide recovery. MS1 intensity values have been applied to profiles of circulating plasma-derived exosomes to improve sensitivity in quantifying protein abundance due to sample input limitations and concomitant sparsity in the MS2 count data.

6) *Same concerns apply to Fig 4B and C. What do the different data in the heat map in Fig 4C correspond to?*

The heatmaps in Figures 4B and C were generated through complete linkage hierarchical clustering of the proteomics data (normalized MS2 counts); Pearson correlation was applied as the basis for distance measure. The heatmap in Figure 4C depicts clustering on proteins identified by unsupervised hierarchical clustering analysis to be (left panel) enriched in the exosome surfaceome (orange bar) relative to total exosome extract (TEE) and cargo compartments and (right panel) on the exosome surface compared to the cell surface. This has now been clarified in the figure legend.

7) *The high level of intracellular proteins found on exosomes is a concern. It could be that simply apoptotic debris aggregates and accumulates non-specifically on endosome surfaces. How confident were these IDs? Were they based on multiple peptides a fact that would enhance the likelihood that the intact protein and not proteolytic fragments somehow ended up on the surface of exosomes.*

We now report in Supplementary Table 7 the list of identified peptides per individual cell line for each PDAC exosome surfaceome protein presented in Figure 5A and Supplementary Table 6. The analysis reveals high sequence coverage, suggesting that intact proteins rather than proteolytic fragments were enriched on the exosome surfaceome. Also, many proteins classically considered as intracellular are nevertheless associated with membranous subcellular compartments or organelles; these could thus be secreted in association with exosomes as a result of tumor-reprogramming of normal intracellular endocytotic sorting and trafficking functions.

8) *Line 279: These data provide evidence of a direct binding of autoantibodies to intact exosomes in contrast to free circulating proteins," I do not see how this conclusion is supported by the data!*

This sentence has now been removed.

Minor:

1) *Line 290; "As B lymphocytes are able to recognize conformational epitopes, our findings suggest that, exosomes must expose B cell epitopes as native antigens." This does not follow in any way from the data. BCRs can recognize either peptides or conformational epitopes!*

This sentence has now been removed.

REVIEWERS' COMMENTS:

Reviewer #1 (Remarks to the Author):

The manuscript is much improved and the authors have satisfactorily responded to all the concerns raised. A few key pieces of new data remain incomplete and are important to assess the updated data and the authors responses in its entirety.

1. Supplementary Tables 2, 3, and 4 are missing.

Reviewer #2 (Remarks to the Author):

I am satisfied with the changes and the comments made, I think the paper should be published.

Reviewer #3 (Remarks to the Author):

I am satisfied with the Authors' response to my comments. One minor remaining question that should be clarified: The authors report CDC inhibition of patient sera by exosomes from PANC-1 or Pa-O3 cells via a decoy mechanism. This implies that most of the proteins in exosomes from cell lines are the same as proteins from patients. Is that the case?

NCOMMS-18-02505B, M. Capello, *et al.*

Point-by-point response to the Reviewer's comments following second review:

Reviewer 1: *The manuscript is much improved and the authors have satisfactorily responded to all the concerns raised.*

We thank the reviewer for acknowledging the improvement of our report.

1. Supplementary Tables 2, 3, and 4 are missing.

We apologize for the confusion regarding these Supplementary Tables. We have confirmed that they are included in the current Supplementary Excel file.

Reviewer #2: *I am satisfied with the changes and the comments made, I think the paper should be published.*

We thank the Reviewer for commending our study and recommending publication.

Reviewer #3: *I am satisfied with the Authors' response to my comments. One minor remaining question that should be clarified: The authors report CDC inhibition of patient sera by exosomes from PANC-1 or Pa-O3 cells via a decoy mechanism. This implies that most of the proteins in exosomes from cell lines are the same as proteins from patients. Is that the case?*

We agree with the Reviewer. The results presented in Figure 6a and 6b demonstrate a significant autoantibody reactivity of patient plasmas compared to healthy controls against PDAC cell-derived exosomes. Moreover, through immunoblot and protein array analysis we identified a number of PDAC cell line exosome-associated antigens showing immunoreactivity against PDAC patient autoantibodies (Fig. 5b, Supplementary Fig. 9 and Supplementary Table 1). Finally, a subset of Ig-bound proteins also expressed in PDAC cell line exosomes were elevated in PDAC patient plasma exosomes relative to matched controls (Fig. 5d). Altogether, these data suggest that antigenic proteins enriched in PDAC cell line exosomes are present in PDAC patient plasma exosomes.